# Molecular mechanism of condensin I activation by KIF4A

Erin E Cutts [ID][1,2✉], Damla Tetiker[3,4], Eugene Kim [ID][3] & Luis Aragon [ID][2✉]

## Abstract

**During mitosis, the condensin I and II complexes compact chromatin into chromosomes. Loss of the chromokinesin, KIF4A, results in reduced condensin I association with chromosomes, but the molecular mechanism behind this phenotype is unknown. In this study, we reveal that KIF4A binds directly to the human condensin I HAWK subunit, NCAPG, via a conserved disordered short linear motif (SLiM) located in its C-terminal tail. KIF4A competes for NCAPG binding to an overlapping site with SLiMs at the N-terminus of NCAPH and the C-terminus of NCAPD2, which mediate two auto-inhibitory interactions within condensin I. Consistently, the KIF4A SLiM peptide alone is sufficient to stimulate ATPase and DNA loop extrusion activities of condensin I. We identify similar SLiMs in the known yeast condensin interactors, Sgo1 and Lrs4, which bind yeast condensin subunit, Ycg1, the equivalent HAWK to NCAPG. Our findings, together with previous work on condensin II and cohesin, demonstrate that SLiM binding to the NCAPG-equivalent HAWK subunit is a conserved mechanism of regulation in SMC complexes.**

**Keywords** Condensin; KIF4A; SLiMs; SMC Complexes; Cell Cycle Regulation
**Subject Categories** Cell Cycle; Chromatin, Transcription & Genomics

See also: Wang et al

## Introduction

Ensuring faithful division of genetic material between daughter cells is a challenge every cell faces during mitosis. Cells must solve two important problems before attempting division; (i) interphase chromosomes are much longer than the dividing cell, hence need to be compacted, and (ii) replication causes entanglement between the newly produced sister chromatids, which must be resolved ahead of anaphase (Sundin and Varshavsky, 1981). To overcome these two issues, cells organise interphase chromatin into compact rod-shaped chromosomes during mitosis, reducing their length sufficiently to help avoid chromosome entrapment during cytokinesis and promoting the disentanglement of sister chromatids arms.

In mammalian cells, the key players driving the transformation of interphase chromatin into mitotic chromosomes are two structural maintenance of chromosome (SMC) family members; condensin and cohesin. Condensin creates DNA loops (Ganji et al, 2018; Kong et al, 2020) that organise around an axis to facilitate chromosome compaction (Gibcus et al, 2018), while cohesin provides chromosome "cohesion" by holding chromatids together (Sonoda et al, 2001; Hauf et al, 2001; Gibcus et al, 2018). Cohesin also has a DNA looping activity, (Davidson et al, 2019; Kim et al, 2019) restricted to interphase (Gibcus et al, 2018), which enables long-range genome interactions (Rao et al, 2017), in contrast to condensin, which functions mainly during mitosis.

Both condensin and cohesin create loops by loop extrusion, where DNA loop size is processively increased in an ATP-dependent manner. This mechanism is conserved through evolution, with all SMC complexes shown to have loop extrusion activity in vitro (Ganji et al, 2018; Kong et al, 2020; Davidson et al, 2019; Kim et al, 2019; Higashi et al, 2021; Pradhan et al, 2023), however a full molecular understanding as to how loop extrusion is achieved and regulated remains elusive. The conserved activity of the condensin and cohesin SMC complexes is likely due to a conserved overall architecture, composed of heterodimers of SMC subunits, a kleisin subunit and two different HAWK (HEAT associated with Kleisin) proteins (Fig. 1A). SMC subunits have ~50 nm long coiled-coil regions, which heterodimerise via a hinge domain at one end of the coiled-coils, and sandwich two ATP molecules between split ATPase domains at the other end. The kleisin binds all components of the complex; its N-terminus associates with one SMC, its C-terminus with the other SMC, and the middle section binds both HAWK proteins (Onn et al, 2007). Current models for loop extrusion mechanism suggest each HAWK subunit helps to define compartments that can hold and exchange DNA (Collier et al, 2020; Shi et al, 2020; Shaltiel et al, 2022; Higashi et al, 2021; Dekker et al, 2023). For condensin, the HAWK bound towards the C-terminus of the Kleisin, Ycg1/NCAPG, is thought to anchor the complex to DNA, with structural studies demonstrating that the Kleisin, Brn1/NCAPH, folds and latches over DNA (Kschonsak et al, 2017). While the other HAWK, Ycs4/NCAPD2, along with the Kleisin and ATPase heads defines another compartment necessary for DNA loop growth (Shaltiel et al, 2022; Lee et al, 2022).

How condensin loop extrusion activity is controlled by the cell cycle is a long-standing question. While previous studies have demonstrated phosphorylation by mitotic kinases such as

[1]School of Biosciences, Faculty of Science, The University of Sheffield, Firth Court, Western Bank, Sheffield S10 2TN, UK. [2]DNA Motors Group, MRC Laboratory of Medical Sciences (LMS), Du Cane Road, London W12 0HS, UK. [3]Max Planck Institute of Biophysics, 60438 Frankfurt am Main, Germany. [4]IMPRS on Cellular Biophysics, Max-von-Laue-Straße 3, 60438 Frankfurt am Main, Germany. ✉E-mail: e.cutts@sheffield.ac.uk; luis.aragon@lms.mrc.ac.uk

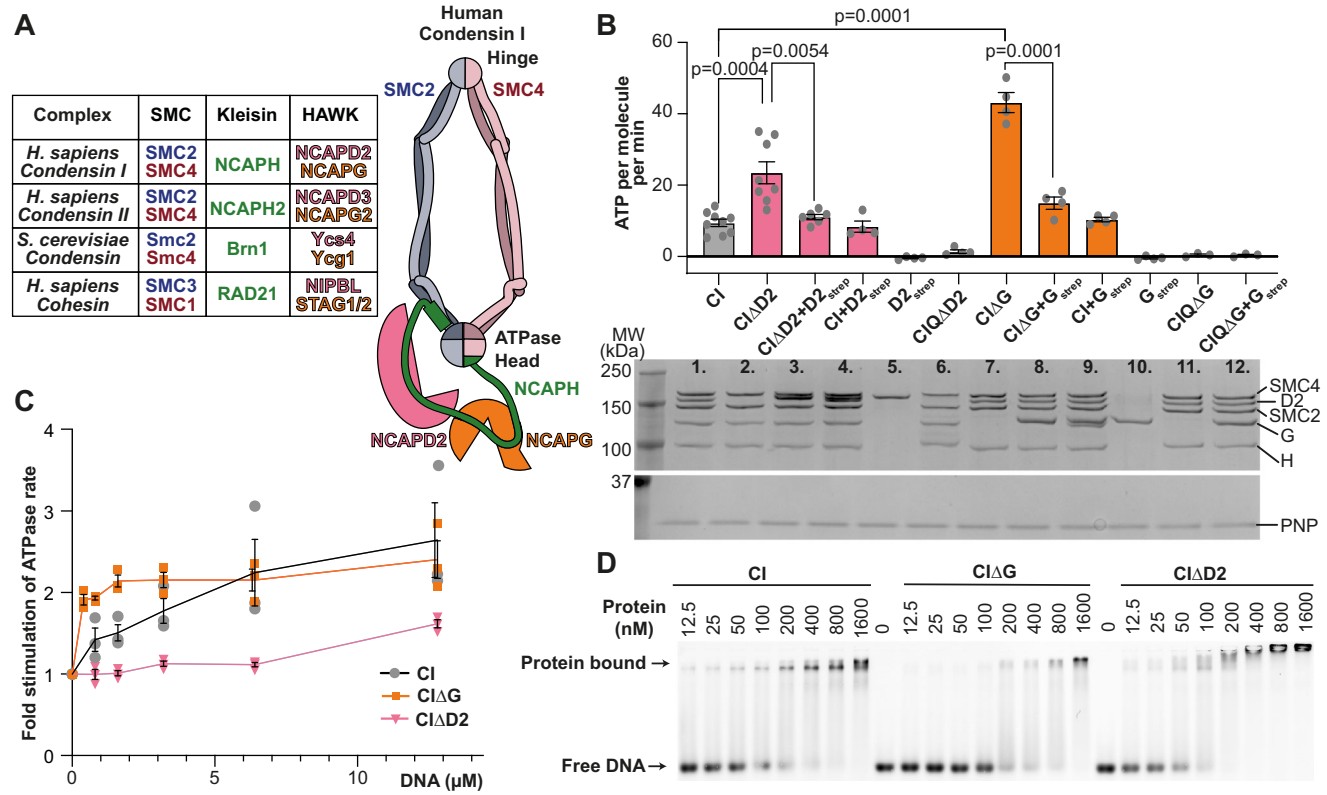

**Figure 1. Condensin I mutations result in increased ATPase activity.**

(**A**) An overview of the subunit architecture of human condensin I, with a table of equivalent subunits in the condensin and cohesin complexes. (**B**) ATPase assay of human condensin I pentamer (CI), tetramers lacking HAWK domains and reconstituted pentamers. Q refers to complexes with a mutation in the Q-loop, which cannot bind ATP. The lower panel shows SDS-PAGE analysis of the ATPase reaction mixture after completion of the assay. Purine nucleoside phosphorylase (PNP) enzyme used in the ATPase assay is indicated as a loading control. Experimental sample sizes for each lane: 1. $n = 9$, 2. $n = 8$, 3. $n = 6$, 4–10. $n = 4$, and 11–12. $n = 3$, where each technical replicate was performed with a distinct protein aliquot at a different time. *P* values determined using unpaired two-tailed *t* tests with Welch's correction to account for unequal variance. (**C**) Fold-stimulation of the ATPase rate when 50 bp dsDNA is titrated, determined by dividing the ATPase rate at each DNA concentration, by the rate without DNA, for each DNA concentration, $n = 3$ technical replicates were performed with a distinct protein aliquot. (**D**) Gel shift assay of condensin I pentamer and tetramers, respectively, binding to 50 bp dsDNA labelled with Cy5. In each case, column value and error bars indicate, mean and s.e.m., respectively. Source data are available online for this figure.

CDK1/cyclin B and Aurora contribute to activation (Bazile et al, 2010), mechanistic detail is lacking. Curiously, recent studies have demonstrated that chromosomes can still be produced by tetrameric condensin I or II lacking the NCAPG or G2 subunit, respectively (Kinoshita et al, 2015; Yoshida et al, 2022), and whilst this work demonstrates tetrameric condensins (lacking NCAPG/2 subunits) are capable of compaction, they lacked finely-tuned regulation of their activity. HAWK subunits NCAPG2 and NCAPG have binding sites for regulatory cofactors. NCAPG2 acts as a binding scaffold for a short linear motif (SLiM) found in MCPH1 and M18BP1, whose binding represses and activates condensin II, respectively (Houlard et al, 2021; Wood et al, 2008; preprint: Borsellini et al, 2024). The chromokinesin, KIF4A, is known to bind NCAPG and promote the association of condensin I to chromosomes as well as the congregation of chromosomes in metaphase (Samejima et al, 2012; Poser et al, 2019; Wang et al, 2020; Takahashi et al, 2016). However, the molecular mechanisms of how KIF4A regulates condensin I activity has not been determined.

In this study, we determine how the condensin I HAWK subunits regulate the core ATPase activity of the complex. We demonstrate that NCAPG acts as a scaffold for three SLiMs; two that cause auto-inhibition, found in the C- and N- terminal disordered regions of NCAPD2 and NCAPH, respectively, and one in KIF4A resulting in activation. Moreover, we show that potential competition between KIF4A and NCAPD2/H binding results in condensin I activation. Importantly, we find SLiM-mediated regulation is a feature that NCAPG shares with other HAWK subunits, such as Ycg1 from yeast condensin, NCAPG2 from condensin II and STAG2 from cohesin.

## Results

### Human condensin tetramers have hyperactive ATPase activity

To determine how HAWK subunits regulate core condensin I activity, tetrameric complexes lacking either NCAPG (CIΔG) or NCAPD2 (CIΔD2) were purified and assayed for ATPase activity, alongside wild-type condensin I pentamer (CI). CIΔG and CIΔD2

hydrolysed ATP 4.6 and 2.5-fold faster, respectively, than CI (Fig. 1B, lanes 2 and 7, compared to lane 1), and this increase in ATPase activity was observed across different purifications of the protein complexes (Fig. EV1A). The hyperactivity of tetrameric complexes was partially rescued towards pentameric levels when an excess of recombinant NCAPG or NCAPD2 were added (Fig. 1B, lanes 3 and 8), with the addition of NCAPG and NCAPD2 to tetrameric complexes resulting in partial reconstitution of pentameric complex (Fig. EV1B,C). The condensin I complex was most stable when all subunits were co-expressed, and while adding HAWK subunits to tetrameric complex partially reconstituted pentamers, complete reconstitution could not be achieved. CIΔD2 and CIΔG with mutations in the Q-loop of ATPase site, unable to bind ATP (CIQΔD2 and CIQΔG), and NCAPD2 or NCAPG alone had negligible ATPase activity (Fig. 1B, lanes 6, 11, 5, and 10, respectively), indicating additional ATPase activity measured is indeed that from the condensin I active site. HAWK subunits were only able to reduce ATP turnover in stoichiometric amounts, as adding excess NCAPG or NCAPD2 to pentameric CI had no effect (Fig. 1B, lanes 4 and 9). Hence, NCAPG and NCAPD2 both have roles in repressing ATPase activity within the condensin I complex.

## NCAPD2 is required for DNA-stimulated ATPase, while NCAPG represses it

The ATPase activity of condensin complexes is known to be stimulated by DNA and structural studies suggest both HAWK domains bind DNA (Lee et al, 2022; Shaltiel et al, 2022; Kschonsak et al, 2017). To determine how each HAWK domain affects DNA-stimulated activity, we performed ATPase assays titrating in 50 bp of double-stranded DNA, and determined the fold-stimulation by dividing the ATPase rate at each DNA concentration by the rate without DNA. Under these conditions, CI ATPase rate is stimulated up to a limit of ~2.4-fold at saturating DNA concentrations (Fig. 1C).

DNA stimulation was largely abolished in CIΔD2 (Fig. 1C), consistent with previous studies of yeast condensin which demonstrate the NCAPD2 equivalent subunit, Ycs4, clamps DNA on top of the ATPase heads to stimulate hydrolysis (Shaltiel et al, 2022). In contrast CIΔG was maximally stimulated at approximately tenfold lower DNA concentrations than pentameric CI, suggesting that NCAPG also represses DNA-stimulated ATPase activity (Fig. 1C).

To ensure the differences in the observed DNA-stimulated ATPase were not due to differences in DNA-binding affinity (Fig. 1C), we examined the binding of tetrameric and pentameric complexes to DNA using gel shift assays. Loss of NCAPD2 resulted in a minimal reduction in binding, and a higher shifted band, suggesting it may multimerise. While, loss of NCAPG resulted in an approximately fourfold reduction in affinity (Fig. 1D), therefore the hypersensitivity of CIΔG to DNA is not due to an increase in DNA-binding affinity but rather due to NCAPG-mediated repression of ATPase activity in response to DNA.

## Mechanisms of NCAPG-derived repression

Our data demonstrates that the presence of HAWK subunits in condensin I complex represses the core ATPase activity (Fig. 1B,C). We hypothesised three potential mechanisms to explain this repression; (1) NCAPG binds to the kleisin via an extended region,

$NCAPH_{421-539}$, hence loss of NCAPG could potentially alleviate structural constraints by lengthening the kleisin, and this might result in higher ATP turnover. (2) A direct interaction between the yeast NCAPG homologue, Ycg1, and SMC2 was observed in the structure of ATP-bound condensin (Lee et al, 2020), raising the possibility that this might inhibit complex activity. (3) Recent work has suggested that disordered regions in non-SMC subunits of condensin I and II can act as regulators of activity during chromosome assembly (Yoshida et al, 2022; Tane et al, 2022), therefore the absence of NCAPG might affect these regulatory regions, resulting in higher ATPase activity.

To investigate these hypotheses, we generated a panel of mutations and deletions (Fig. 2A). First, we generated an internal deletion of $NCAPH_{421-539}$ (CIΔSB) that spans the region of interaction with NCAPG. This deletion did not rescue the increased ATPase rate observed for tetrameric CIΔG (Fig. 2A, lane 4), suggesting that the potential structural constraint on the kleisin imposed upon NCAPG binding is not behind the increased ATPase rate of CIΔG. Furthermore, adding NCAPG to CIΔSB resulted in no change in ATPase rate, indicating that NCAPG must be bound to the kleisin to elicit its inhibitory effect (Fig. 2A, lane 5). Next, we generated mutations in the predicted interface between NCAPG and SMC2 based on the observed Ycg1/SMC2 interface (Lee et al, 2020), NCAPG patch1 and NCAPG patch2. The ATPase rates adding either NCAPG patch1 or patch2 to CIΔG were comparable to wild-type pentameric CI (Fig. 2A, lanes 6 and 7), suggesting that NCAPG/SMC2 interactions do not suppress activity. Finally, we deleted the disordered regions in the C-terminus of NCAPG (CIGΔC, lane 10) or NCAPD2 (CID2ΔC, lane 13), or the N-terminus of NCAPH (CIHΔN, lane 12). While CIGΔC was not significantly different from wild-type, CID2ΔC exhibited a 1.5-fold increase in ATPase activity while CIHΔN resulted in a 2.2-fold increase. Deleting both, CIHΔN,D2ΔC resulted in 2.4-fold increase (Fig. 2A, lane 14). Mass photometry analysis of CID2ΔC, CIHΔN and CIHΔN,D2ΔC confirmed these complexes were pentameric (Fig. EV1D–F) and ATPase active site Q-loop mutations had negligible activity (Fig. 2A, lanes 15–17). These results suggest that the disordered regions in NCAPD2 and NCAPH reduce the core ATPase rate of condensin I.

## Direct interaction between SLiMs in NCAPD2 and NCAPH with NCAPG regulates complex activity

Next, we sought to explore the mechanisms by which the disordered regions of NCAPD2 and NCAPH might affect the ATPase activity. Previous studies have shown that a weak interaction may exist between NCAPD2/D3 and G/G2 (Kong et al, 2020; Onn et al, 2007; Ball et al, 2002). We therefore considered the possibility that interactions between NCAPG and the disordered regions of NCAPD2 and NCAPH could inhibit ATPase activity, particularly as CIΔG exhibited the strongest increase in ATPase rate (Fig. 1B).

We ran AlphaFold2 and 3 multimer predictions of these regions with NCAPG (Jumper et al, 2021; preprint: Evans et al, 2022; Abramson et al, 2024), resulting in prediction of disordered, SLiM based interactions with high-confidence pLDDT scores and low predicted alignment error in both cases (Figs. 2B,E and EV2A–C,E–G). The interface between the NCAPD2 SLiM and NCAPG was composed of a conserved basic region, $NCAPD2_{R1393, R1397, R1398}$, an acidic region, $NCAPD2_{E1378, E1381 \text{ and } E1383}$, and a buried hydrophobic,

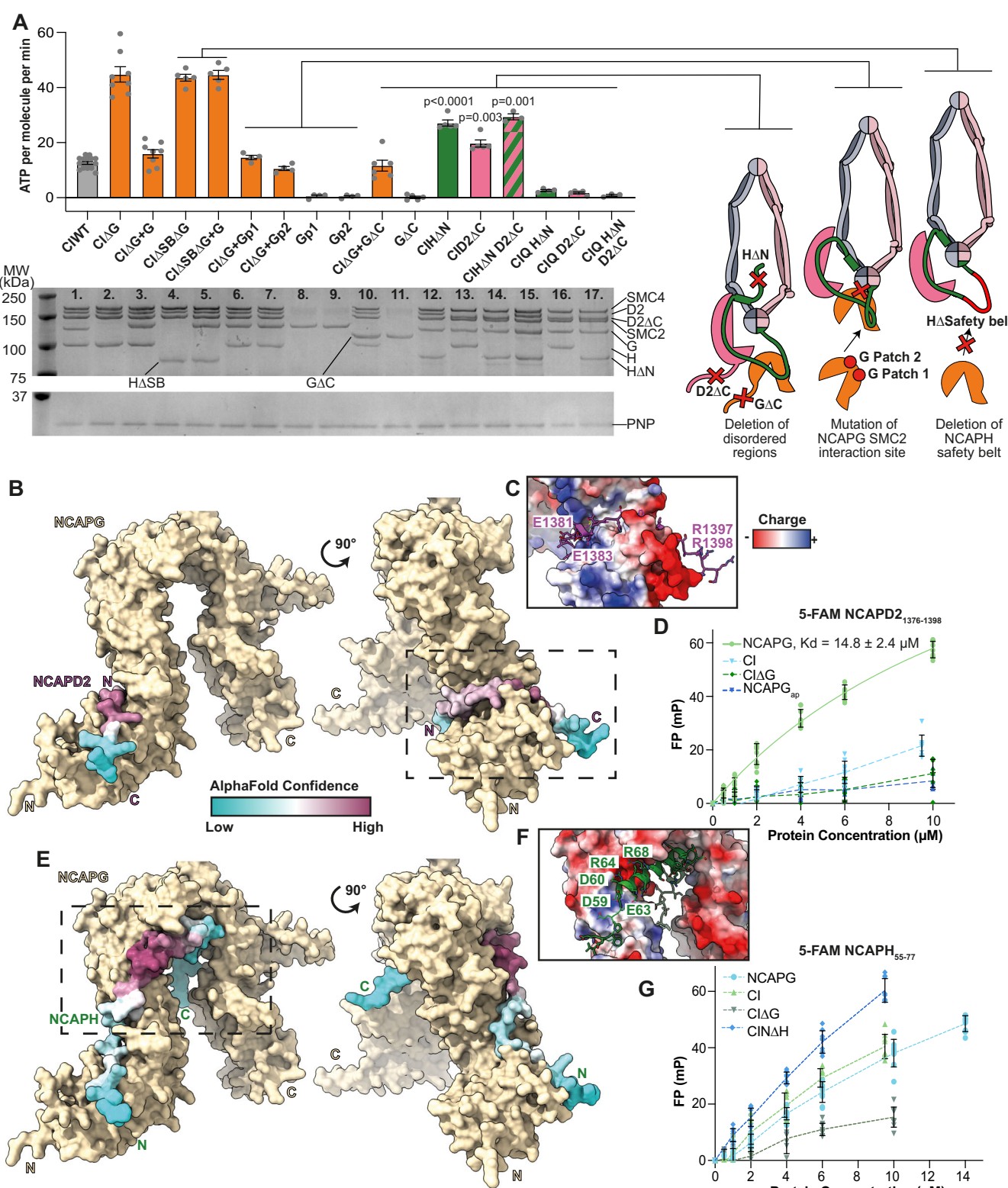

NCAPD2$_{I1391}$ (Figs. 2B,C and EV2A–D). The interface between NCAPH SLiM and NCAPG was distinct from that bound by NCAPD2, and is composited of acidic and basic patches, NCAPH$_{D59, D60, E61, E63}$ and NCAPH$_{R64, R67, R68, R69 and R71}$, respectively (Figs. 2E,F and EV2E–H).

To validate the interactions predicted by AlphaFold2 and 3, we tested whether peptides containing these regions could interact with purified NCAPG and condensin I. To this aim, we used fluorescent polarisation (FP) binding assays using 5-FAM labelled peptides composed of NCAPD2$_{1376–1398}$ or NCAPH$_{55–77}$.

◀ **Figure 2. NCAPG binds disordered SLiMs regions of NCAPD2, NCAPH.**

(A) ATPase assay of condensin I mutants, with mutations illustrated on the right. Mutation CIΔG indicate tetrameric condensin I lacking NCAPG. CIΔSB indicates deletion of the NCAPH region where NCAPG binds. Gp1 and Gp2 indicates mutations in NCAPG homologous to patch1 and patch2 where Ycg1 and SMC2 interact in yeast condensin. CIGΔC indicates deletion of NCAPG C-terminal disordered region, CINΔH indicates deletion of NCAPH N-terminal disordered region and CID2ΔC indicates deletion of NCAPD2 C-terminal disordered region. CIQHΔN, CIQD2ΔC and CIQHΔN,D2ΔC indicating ATPase dead Q-loop mutant controls. The lower panel is SDS page analysis of the ATPase reaction mixture after completion of the assay. Purine nucleoside phosphorylase (PNP) enzyme used in the ATPase assay is indicated as a loading control. Experimental sample size for each condition: 1. $n = 17$, 2–3. $n = 8$, 4–5. $n = 5$, 6–9. $n = 4$, 10. $n = 6$, 11–13. $n = 5$, 14. $n = 3$, 15. $n = 4$, 16. $n = 5$ and 17. $n = 3$, where each technical replicate was performed using a distinct protein aliquot, collected at a different time. $P$ values shown are comparisons to CIWT, determined using unpaired two-tail t tests with Welch's correction to account for unequal variance. In each case, column value and error bars indicate mean and s.e.m., respectively. (B) AlphaFold2 model of NCAPG with the C-terminal end of NCAPD2, with NCAPD2 coloured with AlphaFold2 (Jumper et al, 2021; preprint: Evans et al, 2022) pLDDT confidence score. (C) Surface charge of the NCAPG/D2 interface. (D) Fluorescence polarisation binding assay of 5-FAM NCAPD2$_{1376-1398}$. Error bars indicate mean ± s.d., data from three technical replicates, performed with distinct protein aliquots, read three times. Solid line indicates data fit to determine Kd. "NCAPGap" indicates NCAPG with acid patch mutation (D136K, D137K, D141K). (E–G) Equivalent data as (B–D) for NCAPH. Fluorescence polarisation in (G) was performed with 5-FAM NCAPH$_{55-77}$. Source data are available online for this figure.

5-FAM-NCAPD2$_{1376-1398}$ titrated with NCAPG resulted in an increase in FP signal and a fit Kd of $14.8 ± 2.4\ \mu M$ (mean ± s.e.m.) (Fig. 2D). While, titrating CIΔG or NCAPG with a charge reversal in the acidic patch (NCAPG$_{ap}$) that is bound by NCAPD2$_{R1393, R1397, R1398}$ resulted in minimal binding (Fig. 2D), consistent with specific binding of 5-FAM-NCAPD2$_{1376-1398}$ to NCAPG. Titrating condensin I resulted in less binding than NCAPG alone, potentially suggesting the NCAPD2 peptide was competing with the corresponding region within the condensin I complex (Fig. 2D).

The 5-FAM-NCAPH$_{55-77}$ peptide resulted in a larger increase in FP signal when titrating either NCAPG alone, CI or CINΔH, as compared to CIΔG, suggesting specific binding to NCAPG, albeit with low affinity (Fig. 2G). Collectively, these results support the interactions predicted by AlphaFold.

## Condensin I inhibitory peptides compete with KIF4A

Our results suggest that human condensin I is partially auto-inhibited, hence could be activated in vivo by overcoming these interactions. In cells, condensin I is known to interact with the chromokinesin, KIF4A, promoting condensin I chromosome association (Takahashi et al, 2016; Samejima et al, 2012; Poser et al, 2019). AlphaFold2 and 3 (Jumper et al, 2021; preprint: Evans et al, 2022; Abramson et al, 2024) predicted a high-confidence interaction between NCAPG and a conserved SLiM in the C-terminal tail of KIF4A (Figs. 3A and EV2I–K). The KIF4A interaction with NCAPG was mediated by a conserved basic region, KIF4A$_{K1208-1211, R1212}$ and a buried hydrophobic KIF4A$_{L1214}$, which bound to the same site on NCAPG as NCAPD2 (Fig. 3B,C). The KIF4A interface extended further up NCAPG than NCAPD2, making hydrophobic contacts including KIF4A$_{F1220, F1221}$ shown be required for NCAPG binding (Poser et al, 2019; Takahashi et al, 2016) and a lower confidence interaction involving an acidic region, KIF4A$_{E1228-1230}$ which binds the same region of NCAPG as the acidic patch of NCAPH (Fig. 3B,C). The predicted NCAPG/KIF4A interaction, suggests KIF4A binding partially overlaps with that of NCAPD2 and NCAPH, and that KIF4A may compete with these inhibitory interactions to activate condensin I activity (Fig. 3C).

To validate the AlphaFold models we used FP assays with a 5-FAM-labelled KIF4A$_{1206-1228}$. The 5-FAM-KIF4A$_{1206-1228}$ bound to NCAPG with a Kd of $5.9 ± 0.7\ \mu M$, while minimal binding was detected to CIΔG and NCAPG acidic patch mutant (Fig. 3D). KIF4A had a weaker affinity of $17.7 ± 4.7\ \mu M$ for CIWT which

contains the NCAPD2 and NCAPH SLiMs, but bound with a higher affinity Kd of $2.8 ± 0.2\ \mu M$ to CID2ΔC, suggesting KIF4A competes with NCAPD2 to bind NCAPG.

To further validate that KIF4A competes with NCAPD2 and NCAPH binding to NCAPG, we performed FP competition assays. A fixed 100 nM concentration of 5-FAM-NCAPD2$_{1376-1398}$ with 5 μM NCAPG resulted in a mean FP ± SD of $43 ± 2$ mP, which was significantly reduced to $11 ± 7$ mP on addition of competing KIF4A$_{1206-1228}$ peptide. FP signal was not significantly changed when a "KIF4A mut" peptide harbouring mutations in the interaction interface (K1210A, K1211A, R1212A, L1214G, S1216A, N1217A) was used, demonstrating KIF4A peptide binding competed with 5-FAM-NCAPD2$_{1376-1398}$ probe (Fig. 3E). Similarly, a fixed 100 nM concentration of 5-FAM-NCAPH$_{55-77}$ with CID2ΔC,NΔH results in FP signal of $37 ± 7$ mP, which is significantly reduced to $25 ± 6$ mP on addition of KIF4A$_{1206-1228}$ peptide (Fig. 3F). Collectively, this suggests both NCAPD2 and NCAPH compete with KIF4A to bind NCAPG.

## KIF4A activates condensin I ATPase activity

As KIF4A competes with auto-inhibitory interactions which bind NCAPG, we hypothesised that KIF4A might activate condensin I ATPase activity. We assayed the ATPase activity of CI in the presence and absence of the KIF4A$_{1206-1228}$ peptide and found a small but significant increase in ATPase activity. As KIF4A bound CID2ΔC with higher affinity, we repeated the ATPase assay using CID2ΔC and found a similar increase in ATPase activity on addition of the KIF4A$_{1206-1228}$ peptide. Importantly addition of the KIF4A mut peptide, which does not bind NCAPG, resulted in no increase in activity (Fig. 3G). We found no additional stimulation on ATPase rate when adding KIF4A$_{1206-1228}$ to either CINΔH or CINΔH,D3ΔC, suggesting that KIF4A activates condensin I by competing and reducing binding of the inhibitory N-terminal region of NCAPH. Collectively, this suggests a model where NCAPD2 alters KIF4A binding affinity, and KIF4A competes with NCAPH to enhance ATPase activity.

## KIF4A promotes condensin I loop initiation

As KIF4A was able to activate condensin I ATPase activity, and has previously been shown to promote chromosomal localisation of condensin I (Poser et al, 2019), we determined the effect KIF4A had

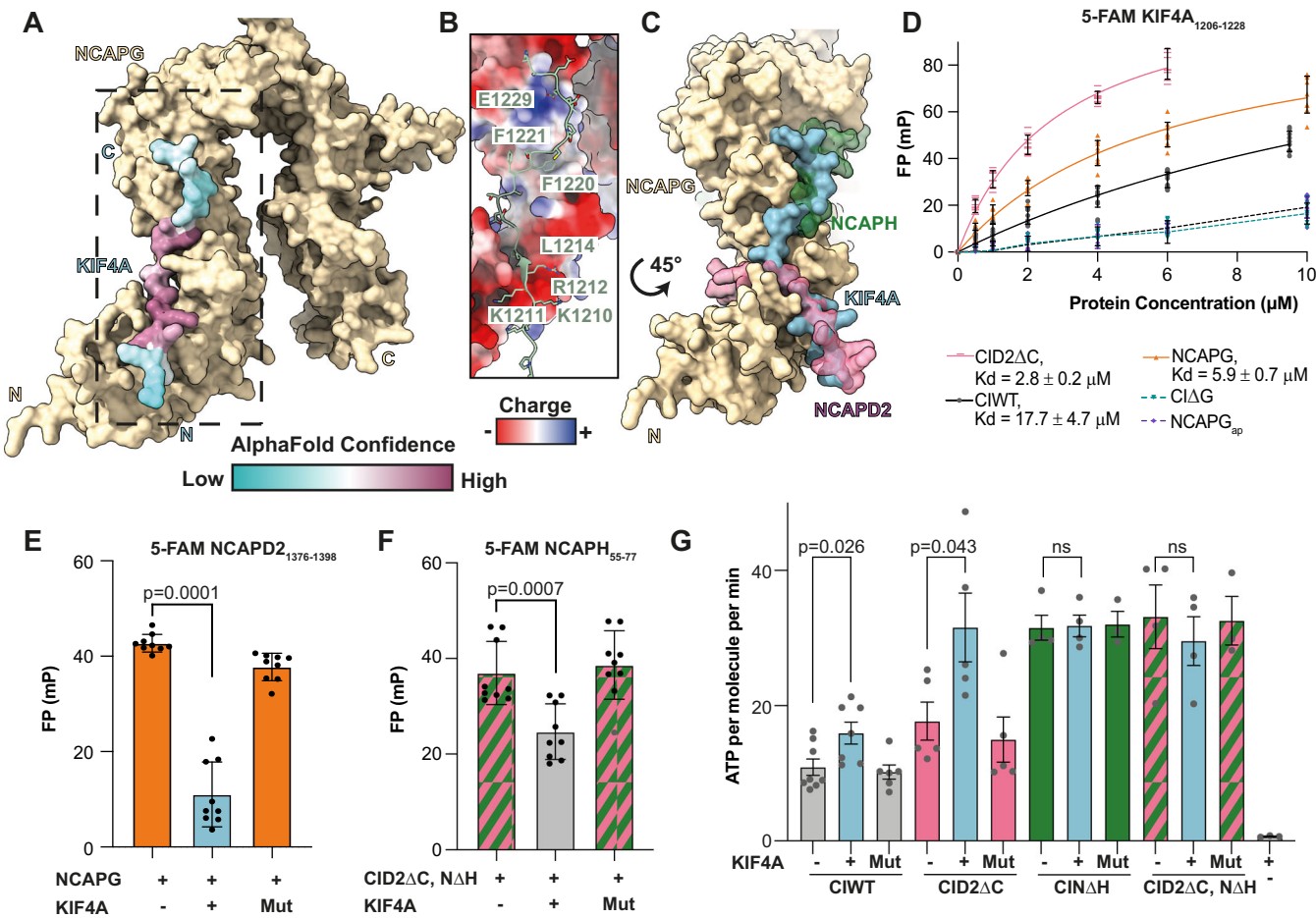

**Figure 3. NCAPG binds disordered SLiMs regions of KIF4A.**

(A) AlphaFold2 model of NCAPG with the C-terminal end of KIF4A, with KIF4A coloured with AlphaFold2 (Jumper et al, 2021; preprint: Evans et al, 2022) pLDDT confidence score. (B) Surface charge of the NCAPG/KIF4A interface. (C) Overlay of NCAPG showing binding of KIF4A peptide (blue) overlaps with binding of NCAPD2 and NCAPH (pink and green, respectively). (D) Fluorescence polarisation binding assay of 5-FAM labelled KIF4A. "NCAPGap" indicates NCAPG with acid patch mutation (D136K, D137K, D141K). Solid line indicates Kd fit, Kd value shown ± standard error. (E) Competition FP assay with 100 nM 5-FAM NCAPD2$_{1376-1398}$, 6 μM NCAPG with 10 μM KIF4A, KIF4A$_{1206-1228}$, WT or Mut (mutant) peptide. (F) Competition FP assay with 100 nM 5-FAM NCAPH$_{55-77}$, 4 μM CINΔH,D3ΔC with 100 μM KIF4A, KIF4A$_{1206-1228}$, WT or Mut peptide. For FP data in (D–F), error bars indicate mean ± s.d., and data were derived from three technical replicates performed with distinct protein aliquots, read three times. *P* values determined using unpaired, two-tailed *t* tests. (G) ATPase assay of WT and pentameric condensin I lacking the C-terminal region of NCAPD2 and/or the N-terminal region of NCAPH, in the presence and absence of 50 μM KIF4A$_{1206-1228}$, WT or Mut peptide. ATPase assays from 3 repeats, error bars indicate s.e.m., *P* values determined with unpaired, two-tailed *t* test with Welch's correction. Source data are available online for this figure.

on condensin I loop extrusion. We used a well-established single-molecule assay for directly visualising loop extrusion by SMC complexes, whereby a 48.5 kbp piece of lambda DNA is flow stretched, attached to a passivated surface via streptavidin–biotin binding and stained with Sytox Orange (SxO) for fluorescence imaging (Ganji et al, 2018). DNA is visualised with total internal reflection microscopy (TIRF) and can be stretched perpendicular to its attachment axis by buffer flow (Fig. 4A). After addition of 1 nM of condensin I and 2.5 mM ATP under flow a DNA punctum forms, which gradually increases in size over time confirming the event as DNA loop extrusion (Fig. 4A; EV Movie 1).

First, we tested CIΔG, and found it was able to create loops. However, the proportion of tethers looped was less than for CIWT (Fig. 4B), likely due to reduced DNA-binding affinity (Fig. 1D). Next, we repeated the assay in the presence or absence of 0.5 μM of

KIF4A$_{1206-1228}$ peptide, reducing the condensin I concentration to 0.5 nM. While CIWT looped ~24% of tethers, addition of KIF4A$_{1206-1228}$ resulted in significantly more loops, with ~96% of tethers having DNA loops, suggesting KIF4A promoted loop formation (Fig. 4C).

We quantified the loop extrusion rate in the absence of flow (EV Movie 2) by measuring how the length of DNA above and below the loop (up and down) changed over time (Fig. 4D–F). While CIΔG increased loop extrusion rate compared to the CIWT, addition of KIF4A$_{1206-1228}$ peptide resulted in no significant difference (Fig. 4G). This suggests KIF4A may promote loop initiation rather than affecting loop extrusion rate. Current models of loop extrusion (Shaltiel et al, 2022; Dekker et al, 2023) suggest that loop initiation requires DNA to be "threaded" through compartments formed by the kleisin (Fig. 5A). Interactions

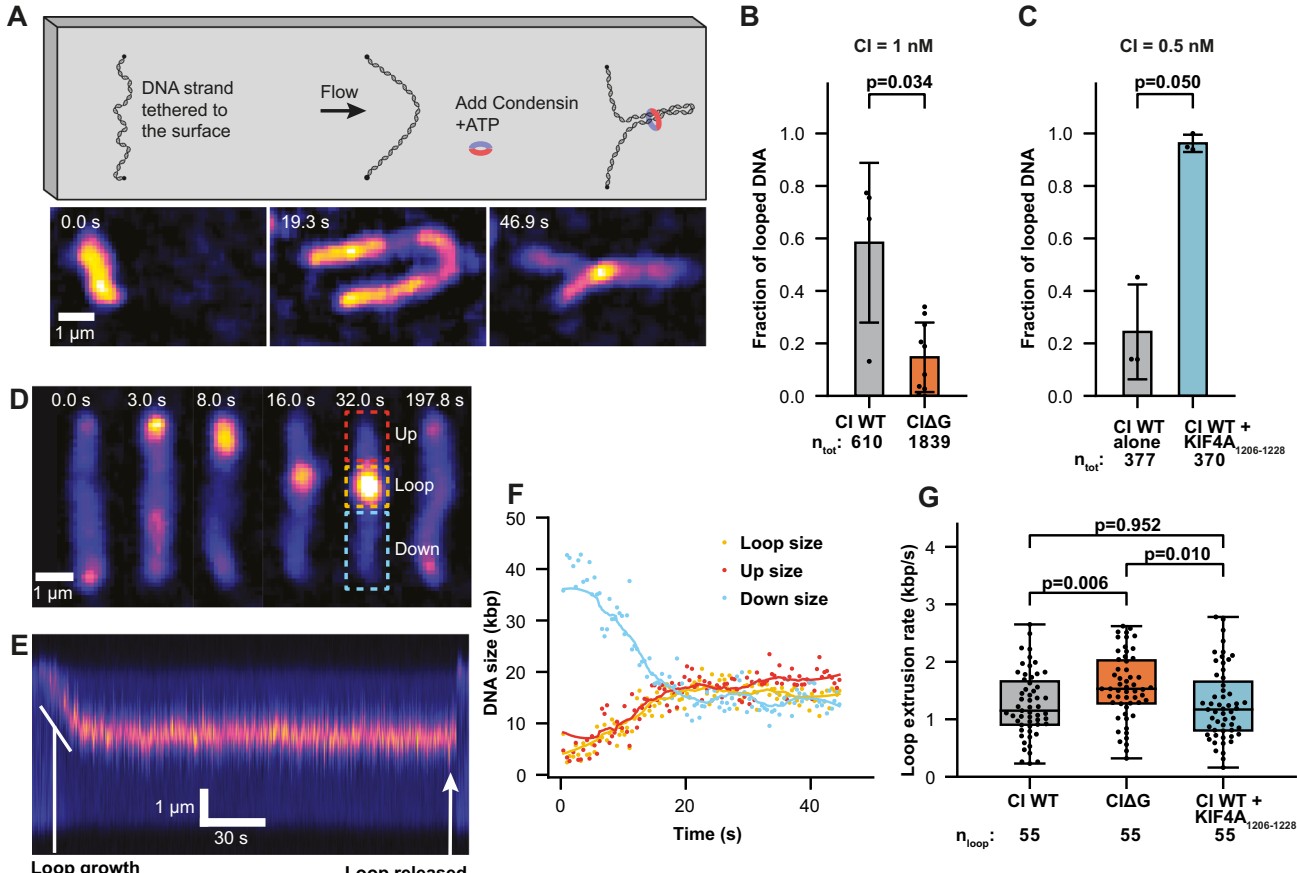

**Figure 4.  KIF4A increases condensin I loop initiation.**

(A) Schematic of the single-molecule loop extrusion assay under constant buffer flow and corresponding snapshots of a loop formed by CIWT in the presence of KIF4A$_{1206-1228}$ peptide. (B) Fraction of DNA strands with at least one looping event during a 1000 s acquisition time, in the presence of 1 nM human CIWT and CIΔG ($n = 4$ and 10 experiments, respectively). (C) Fraction of DNA strands with at least one looping event during a 1000 s acquisition time, in the presence of 0.5 nM condensin I WT and WT supplied with 1000x KIF4A$_{1206-1228}$ peptide ($n = 3$ experiments). (B, C) Mean ± s.d. is shown, the total number of DNA tethers analysed is indicated by $n_{tot}$ and $P$ values calculated with Wilcoxon rank-sum tests. (D) Example snapshots, the corresponding (E) kymograph and (F) time trace of DNA lengths for a loop extrusion event by CIΔG without flow. (G) Box-whisker-plots showing the loop extrusion rates of CIWT, CIΔG and CIWT in the presence of KIF4A$_{1206-1228}$ peptide. Calculated from $n_{tot} = 55$ looping events, from replicates detailed in (B, C). For the box plots, the central line denotes the median, the box limit denotes the 25th–75th percentile and the whiskers denote minimum and maxima. The $P$ values were calculated using Welch's $t$ test. Source data are available online for this figure.

between the NCAPH/D2 with NCAPG could impede this threading process, inhibiting loop initiation (Fig. 5B). Binding of KIF4A would remove this blockage, allowing DNA threading and condensin I loop initiation (Fig. 5C). Loss of NCAPG could also promote DNA threading, as it completely abolishes inhibitory interactions between NCAPG and NCAPD2/H, however, this complex could have reduced DNA affinity and slippage of the DNA anchor (Fig. 5D).

## Motif binding is conserved across species

We have shown that NCAPG acts as a scaffold, binding multiple SLiMs. The equivalent subunits in condensin II and cohesin, NCAPG2 and STAG1/2, respectively, have also previously been shown to act as a scaffold for SLiM interactions (Houlard et al, 2021; Li et al, 2018; García-Nieto et al, 2023; preprint: Borsellini et al, 2024; Yan et al, 2024; Yuan et al, 2024; Shintomi and Hirano, 2009), hence we speculate that this is a conserved mechanism.

Previous work on *C. thermophilum* condensin showed that tetrameric condensin lacking the NCAPG equivalent, Ycg1, resulted in a higher ATPase rate (Shaltiel et al, 2022). We examined sequence data and found *C. thermophilum* NCAPH equivalent, Brn1, contained the N-terminal motif (Fig. EV3A), suggesting this was conserved outside metazoans.

Although the C-terminal region of NCAPD2 is absent in yeast, NCAPG residues at the interaction site are conserved in yeast, Ycg1 (Fig. EV3B,C), suggesting it may act as binding site for other factors. We searched the literature for factors that have been reported to interact with yeast condensin and performed predictions with AlphaFold2 and 3. Two proteins, Lrs4 and Sgo1, were predicted to bind Ycg1 via a similar SLiM as to that found in NCAPD2/KIF4A (Figs. 6A–C and EV3D–I). FP binding assays indicated that 5-FAM-Lrs4$_{317-337}$ and 5-FAM-Sgo1$_{503-523}$ peptides bound to pentameric yeast condensin with Kds of $12.9 \pm 2.5\,\mu M$ and $4.0 \pm 0.3\,\mu M$, respectively, while displayed negligible binding to condensin lacking Ycg1 (Fig. 6D,E), consistent with the interactions

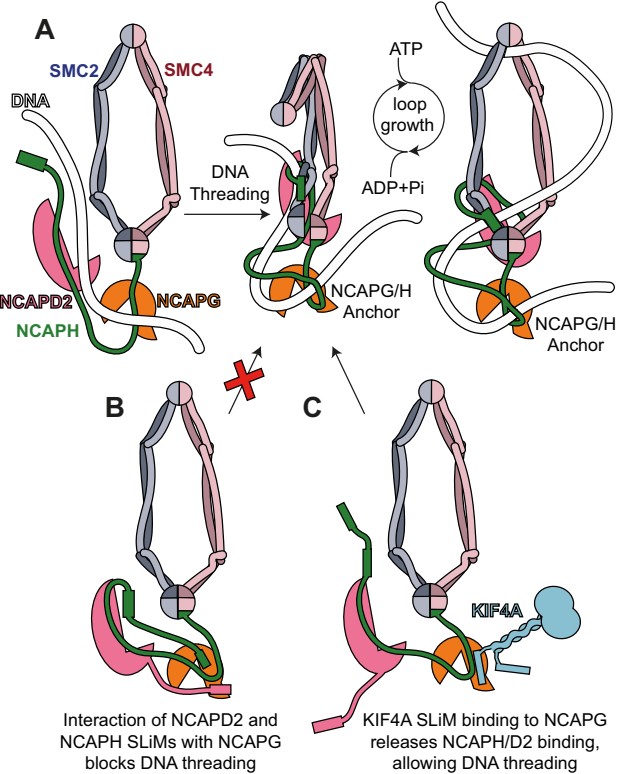

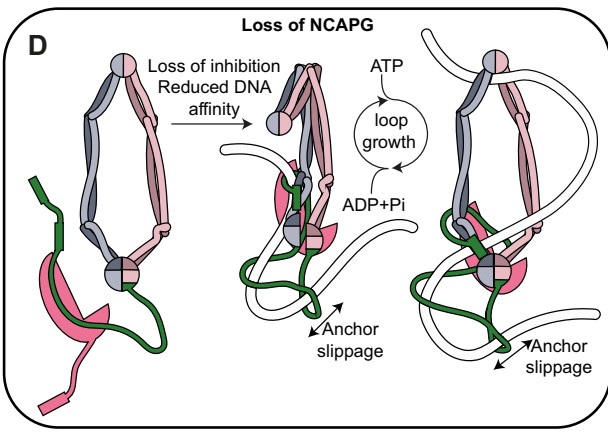

**Figure 5. Model of KIF4A condensin I activation mechanism.**

(A) Current models (Shaltiel et al, 2022; Dekker et al, 2023) for DNA loop extrusion by condensin complexes require loops to be initiated by DNA being "threaded" through the NCAPH Kleisin. (B) In the presence of interactions between NCAPD2/H and NCAPG, the threading step could be blocked. (C) The interaction with KIF4A could allow DNA threading and loop initiation. (D) Loss of NCAPG could also reduce inhibition of DNA threading, but may result in reduced DNA-binding affinity and slippage of NCAPG/H DNA anchor.

predicted by AlphaFold. These results are consistent with recent work demonstrating Sgo1$_{503-523}$ is important for pericentromere enrichment of condensin in *S. cerevisiae* (Wang et al, 2025), and as well as with studies mapping Sgo1/condensin interaction using peptide arrays, in which this region interacted in addition condensin binding to the Serine-Rich Motif at Sgo1$_{137-163}$ (Yahya et al, 2020). Previous work demonstrated Lrs4 with a Q325stop

mutation, which would lack the condensin interaction region, exhibited shortening of rDNA (Johzuka and Horiuchi, 2009), suggesting that an interaction between Lrs4$_{317-337}$ and condensin is required for rDNA stability. Collectively, these findings provide support for the physiological significance of Ycg1/SLiM interactions and a general mechanism of regulation by SLiMs binding HAWKs (Fig. 6F).

## Discussion

KIF4A has long been implicated in chromosome formation, with cellular studies demonstrating that disruption of condensin I/KIF4A association results in reduced condensin I loading, poorly condensed chromosomes and congregation defects (Poser et al, 2019; Samejima et al, 2012). Our work provides a potential molecular mechanism to explain this phenotype, where KIF4A activates the condensin I complex by competing off auto-inhibitory interactions to promote what we term "DNA threading" (Fig. 5). Our data suggests that KIF4A binding affinity is modulated by NCAPD2, while KIF4A competition with NCAPH results in an increase in condensin I activity. Binding of NCAPD2 to NCAPG could contribute to the affinity of NCAPH to NCAPG, as CID2ΔC also has increased ATP turnover (Fig. 2A). We hypothesise that inhibition is due to the fact that interactions between NCAPH/D2 and NCAPG promote a conformation less able to hydrolyse ATP and load onto DNA (Fig. 5). An alternative hypothesis is that interaction between NCAPH and NCAPG alters binding of the N-terminus of NCAPH to the SMC2 neck. Release of the N-terminal Kleisin from the SMC neck has been shown to occur during the ATPase cycle in yeast condensin and cohesin (Hassler et al, 2019; Muir et al, 2020), hence altering the stability of this interaction may affect ATPase rate.

Condensin I activation has previously been attributed to phosphorylation events, and while this work suggests they are not solely responsible, they likely contribute. The SLiMs of NCAPH, NCAPD2 and KIF4A have numerous phosphorylation sites (Fig. EV4A). Three kinase families are likely involved. Firstly, phosphorylation sites have been identified in NCAPD2$_{1366-1386}$, which are enriched in G1 (Dephoure et al, 2008). The sequence in this region suggests acidic directed kinases, such as casein kinases, which have previously been suggested to phosphorylate NCAPD2 and inhibit condensin I during interphase (Takemoto et al, 2006). Secondly, CDK1/cyclin B activity peaks during mitosis and all three SLiMs identified contain canonical S/TP CDK1 consensus motifs (Songyang et al, 1994). Finally, Aurora kinases have been implicated in KIF4A-mediated condensin I activity. Inhibition of Aurora B kinase and mutation of the Aurora consensus site in the NCAPH SLiM, S70A, reduces KIF4A and condensin I chromosome association (Poonperm et al, 2017), while Aurora A activity is essential for KIF4A-dependent chromosome congregation (Poser et al, 2019). Aurora kinases have been demonstrated to phosphorylate both NCAPH$_{S70}$ and NCAPD2$_{S1395}$ (Lipp et al, 2007; Sardon et al, 2010) (Fig. EV4A). Based on the AlphaFold2 models, the surface charge of NCAPG suggests phosphorylation by casein kinase could enhance NCAPD2 association, aiding condensin I repression in G1, while CDK1 and Aurora phosphorylation would result in charge clashes for inhibitory NCAPD2/NCAPH interactions, while adding complementary charge for the KIF4A

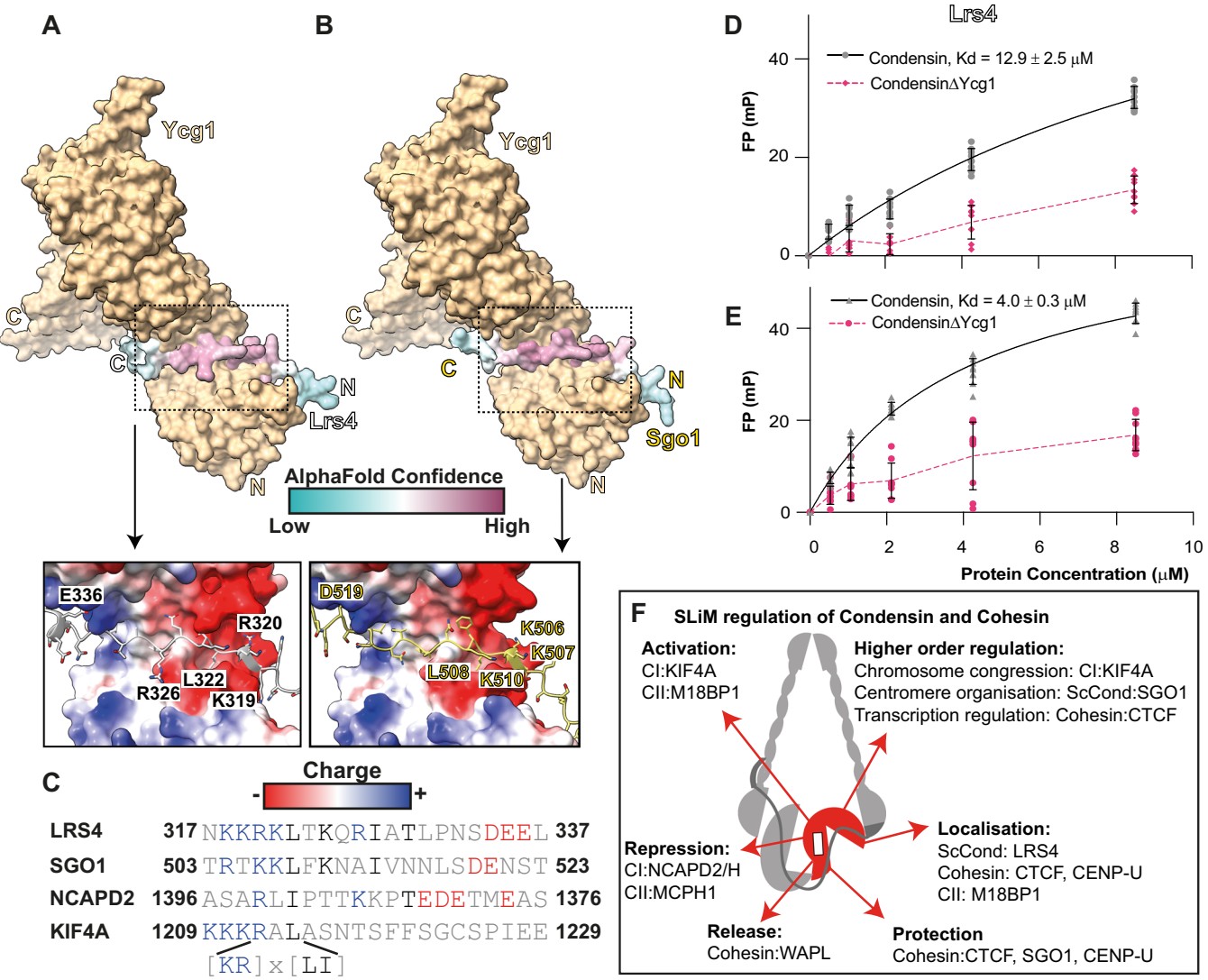

**Figure 6.  SLiM HAWK association is a general mechanism in condensin and cohesin.**

(**A, B**), AlphaFold2 model of Ycg1 with Lrs4 and Sgo, respectively, with insert of electrostatic potential. (**C**) Sequence alignment of SLiMs that bind the equivalent site of NCAPG and Ycg1. (**D, E**) Fluorescence polarisation binding data of Lrs4 and Sgo peptides with yeast condensin complex. Error bars indicate mean ± s.d., data from three technical replicates performed with distinct protein aliquots, read three times. (**F**) A general model of the role SLiM HAWK association has in regulating condensin and cohesin activity. Source data are available online for this figure.

interaction, potentially aiding binding of KIF4A in mitosis. Adding to the complexity of regulation by phosphorylation is the role of phosphatases, as KIF4A and NCAPD2 have binding motifs for protein phosphatase 2 A:B56 and 4, respectively (Wang et al, 2020; Ueki et al, 2019). Future studies should address the effects and timings of phosphorylation on the spatial and temporal regulation of condensin I activity.

Our findings complement prior studies showing that NCAPG deletion permits chromosome formation, while deletion or phospho-mimetic mutations of NCAPH's N-terminus increases loop extrusion and chromosome condensation (Kinoshita et al, 2015; Tane et al, 2022). Our observed loop extrusion rate of 1.2–1.5 kbp/s is

comparable to that previously determined for human condensin I (Tane et al, 2022; Kong et al, 2020). However, the addition of the KIF4A peptide results in a greater enhancement of the fraction of tethers with loops on than deletion of NCAPH N-terminus (Tane et al, 2022), possibly due to KIF4A competing with both the NCAPH and NCAPD2 SLiMs. Our data indicates that the binding affinity of KIF4A to condensin I is relatively weak. Previous studies demonstrated that the cysteine-rich region of KIF4A is essential for condensin I binding (Poser et al, 2019), suggesting an additional KIF4A/condensin I interaction site, which could result in binding avidity, hence higher affinity. However, fluorescence recovery experiments indicate that KIF4A is more dynamic on chromosomes

than condensin I (Samejima et al, 2012), suggesting KIF4A and condensin I could form a transient complex in cells.

Knockdown of both NCAPG and NCAPG2 results in loss of chromosome compaction, suggesting condensin tetramers are not active in human cells (Ono et al, 2003). This indicates an essential role for NCAPG, further demonstrated by its embryonic lethality when knocked out (Sun et al, 2022). Our data suggests NCAPG could play an important role as an interaction scaffold enabling site-specific localisation and regulation, and recent work has found that mutation of the SLiM NCAPH$_{58-68}$ which binds NCAPG affects cellular fitness (preprint: Ambjørn et al, 2024). However, condensin tetramers are thought to exist in some eukaryotes, as NCAPG2 gene is absent in Drosophila melanogaster (King et al, 2019) and previous studies demonstrate tetrameric condensin complex can extrude DNA loops but are prone to slippage (Shaltiel et al, 2022).

Our secondary finding is the emerging pattern that the SMC complexes, condensin and cohesin, are finely regulated by the binding of SLiMs to the NCAPG-equivalent HAWK subunits (Fig. 6F). In this study, we demonstrate that SLiMs from condensin I subunits NCAPD2 and H repress complex activity, while a SLiM from KIF4A activates it. SLiM-mediated regulation at this site is conserved across species, with the N-terminal NCAPH SLiM being present in the *C. thermophillium*. This explains the previously observed result that *C. thermophillium* condensin lacking NCAPG-equivalent HAWK, Ycg1, exhibits both higher ATPase activity and a higher proportion of DNA tethers being compacted in loop extrusion assays (Shaltiel et al, 2022). We have identified a conserved SLiM which binds Ycg1 in the known yeast condensin binding partners Sgo1 and Lrs4. The Sgo1 SLiM has been demonstrated to help facilitate recruitment of condensin to pericentromeres (Wang et al, 2025). The Lrs4 SLiM likely helps recruitment of condensin to rDNA, where deletion of the SLiM results in shortening of rDNA (Johzuka and Horiuchi, 2009). The conservation of the Sgo1 and Lrs4 SLiMs is poor outside of *S. cerevisiae*, but the SLiM is found in other proteins suggesting other potential interaction partners (Wang et al, 2025). Whether yeast condensin SLiMs also regulate DNA loop extrusion is an open question, and should be investigated in future in the context of the larger complexes formed by Sgo1 and Lrs4 and the regions of pericentromeric and rDNA they act upon.

SLiM-mediated regulation is also found in condensin II and cohesin. During interphase, condensin II is repressed by MCPH1, which binds via a SLiM to the NCAPG equivalent, NCAPG2 (Houlard et al, 2021). Condensin II is activated in mitosis by a SLiM found in M18BP1, which competes with that found in MCPH1 (preprint: Borsellini et al, 2024). In cohesin, the NCAPG-equivalent subunit, STAG1/2, is bound by SLiMs found in WAPL, promoting cohesin unloading, CTCF, resulting in stalling/localisation, CENP-U, promoting centromere cohesion, and SGO1, protecting centromeric cohesin (Li et al, 2020; García-Nieto et al, 2023; Yan et al, 2024; Yuan et al, 2024; Shintomi and Hirano, 2009). In many of these examples, SLiM interactions are accompanied by other interactions, resulting in avidity, which is a common feature of SLiM interactions (Holehouse and Kragelund, 2024). Hence, this work and the work of others demonstrate that SLiMs binding HAWK subunits of the SMC complexes condensin and cohesin is a conserved mechanism, resulting in diverse regulatory outcomes (Fig. 6F).

# Methods

**Reagents and tools table**

| Reagent/resource | Reference or source | Identifier or catalogue number |
|---|---|---|
| **Recombinant DNA** | | |
| pBIG2abc Condensin I strep | Kong et al, 2020 | |
| pBIG2abc Condensin I Q strep (SMC2 Q147L, SMC4 Q229L) | Kong et al, 2020 | |
| pBIG2abc Condensin I delG strep (Lacking NCAPG) | This work | |
| pBIG2ab Condensin I delD2 strep (Lacking NCAPD2) | This work | |
| pBIG2ab Condensin I Q delG ystrep (Lacking NCAPG, SMC2 Q147L, SMC4 Q229L) | This work | |
| pBIG2ab Condensin I Q delD2 ystrep (Lacking NCAPD2, SMC2 Q147L, SMC4 Q229L) | This work | |
| pLIB NCAPG ystrep | This work | |
| pLIB NCAPD2 ystrep | This work | |
| pLIB NCAPG delC ystrep (Deletion after residue 913) | This work | |
| pLIB NCAPD2 delC ystrep (Deletion after residue1305) | This work | |
| pLIB NCAPG patch1 ystrep (Mutation: D413K, E416K, E417K, R420D, K421D) | This work | |
| pLIB NCAPG patch2 ystrep (Mutation: Q460A, E464K, S467A, E468K) | This work | |
| pLIB NCAPG acidic patch (G$_{ap}$, mutation D136K, D137K, D141K) | This work | |
| pBIG2ab Condensin I NCAPH delSB strep (NCAPH deletion of residues 421–539) | This work | |
| pBIG2ab Condensin I delN NCAPH strep (NCAPH deletion residues 1–77) | This work | |
| ScCondensin Pentamer | Lee et al, 2020; Martínez-García et al, 2022 | |
| ScCondensin Tetramer, lacking Ycg1 | Lee et al, 2020; Martínez-García et al, 2022 | |

| Reagent/resource | Reference or source | Identifier or catalogue number |
|---|---|---|
| **Oligonucleotides and other sequence-based reagents** | | |
| 5-FAM-KIF4A WT$_{1206-1228}$<br>5-FAM-PGKKKKRALASNTSFFSGCSPIE | Genscript | |
| KIF4A WT$_{1206-1228}$<br>PGKKKKRALASNTSFFSGCSPIE | Genscript | |
| KIF4A Mut$_{1206-1228}$<br>PGKKAAAAGAAATSFFSGCSPIE | Genscript | |
| 5-FAM NCAPD2-C$_{1376-1398}$<br>5-FAM-SAEMTEDETPKKTTPILRASARR | Genscript | |
| 5-FAM N-NCAPH$_{55-77}$<br>5-FAM-FPQNDDEKERLQRRRSRVFDLQF | Genscript | |
| 5-FAM Lrs4$_{317-339}$<br>5-FAM-NKKRKLTKQRIATLPNSDEEL | Genscript | |
| 5-FAM Sgo1$_{503-523}$<br>TRTKKLFKNAIVNNLSDENST | Genscript | |
| 50nt Forward<br>GGTGTGACAGGGTGTGACAGGGTGTGACAGGGTGTGACAGGGTGTGACAG | IDT | |
| Cy5 50nt Reverse<br>/5Cy5/CTGTCACACCCTGTCACACCCTGTCACACCCTGTCACACC | IDT | |
| 50nt Reverse<br>CTGTCACACCCTGTCACACCCTGTCACACCCTGTCACACCCTGTCACACC | IDT | |
| **Chemicals, enzymes and other reagents** | | |
| Gibson Assembly Master Mix | NEB | E2611 |
| DH10EMBacY cells | Geneva biotech | |
| Sf9 Insect Cells | Novagen | 71104 |
| Cellfectin II reagent | Gibco | 10362100 |
| Refeyn sample carrier slide | Refeyn | MP-CON-21009 |
| NativeMark | Thermo Scientific | LC0725 |
| EnzCheck phosphate assay kit | Invitrogen | E6646 |
| Double-stranded (48502 bp) Lambda Phage DNA | NEB | N3011L |
| **Software** | | |
| GraphPad Prism | https://www.graphpad.com/ | |
| AlphaFold2 collabfold | https://colab.research.google.com/github/deepmind/alphafold/blob/main/notebooks/AlphaFold.ipynb<br>https://colab.research.google.com/github/sokrypton/ColabFold/blob/main/AlphaFold2.ipynb | |
| AlphaFold3 webserver | https://alphafoldserver.com | |
| Python package Napari | https://github.com/Napari/napari | |
| Python package PyQtGraph | https://github.com/pyqtgraph/pyqtgraph | |
| Refeyn AcquireMP | https://www.refeyn.com/ | |
| Refeyn DiscoverMP | https://www.refeyn.com/ | |
| **Other** | | |
| Refeyn TwoMP | Refeyn | |
| POLARstar Omega plate reader | BMG | |

## Protein purification

Constructs used for expression are listed in the Reagents and Tools table. Mutations of NCAPH, NCAPD2 and NCAPG were generated using PCR, gel extracted and cloned using Gibson assembly (NEB, Gibson Assembly Master Mix, E2611). Incorporation of mutation was confirmed with Sanger Sequencing (Genewiz).

Human condensin pentamers and tetramers were purified as previously described in (Kong et al, 2020; Houlard et al, 2021). Briefly, wild-type and mutation of pentamer and tetramer and individual subunits NCAPG and NCAPD2 were cloned into biGBac vectors shown in the Reagents and Tools table. Bacmids were generated by Tn7 transposition in DH10EMBacY cells (Geneva biotech), which were transfected into Sf9 cells with

Cellfectin II (GIBCO) to generate baculovirus. Virus was further amplified in Sf9s cells and each construct was expressed in ~500 mL using either Sf9 insect cells, harvesting 72 h after infection. Cells pellets were lysed in human condensin purification buffer (20 mM HEPES pH 8, 300 mM KCl, 5 mM MgCl$_2$, 1 mM DTT, 10% glycerol) supplemented with 1 Pierce protease inhibitor EDTA- free tablet (Thermo Scientific) or 1 cOmplete protease inhibitor EDTA-free tablet (Roche) per 50 mL and 25 U/ml of Benzonase (Sigma) with a Dounce homogenizer followed by brief sonication. Cleared lysate was loaded onto a StrepTrap HP or StrepTrap XT (Cytiva), washed with purification buffer before being eluted with purification buffer supplemented with 10 mM desthiobiotin or 50 mM Biotin (Sigma), respectively. Eluted fractions were diluted approximately twofold, before being loaded onto a Heparin HiTrap (cytiva) equilibrated with Heparin buffer A (20 mM HEPES pH 8, 5% glycerol, 0.5 mM DTT) with 150 mM NaCl and eluted either via a gradient or step elution using Heparin buffer B (Heparin buffer A supplemented with 1 M NaCl). Size exclusion chromatography was performed using human purification buffer on a Superose 6 16/60 (cytiva) for pentamers or tetramers or Superdex 200 30/10 column (cytiva) for individual HAWK protein. Protein-containing fractions separated from the void volume were pooled, concentrated and flash-frozen.

Pentamers with HAWK mutations were co-expressed either from one virus or by co-infection of baculo viruses for condensin I tetramers and HAWK subunit mutation.

Yeast condensin pentameric and tetrameric complex were purified as described previously (Martínez-García et al, 2022; Lee et al, 2020).

## Mass photometry

Mass photometry using a Refeyn TwoMP was performed to demonstrate pentamer reconstitution and confirm the mass and homogeneity of mutations. A Refeyn sample carrier slide (Refeyn, MP-CON-21009) was mounted on the sample stage with a fresh silicon CultureWellTM gaskets (GBL103250, Sigma-Aldrich) attached to centre. All samples were measured in 20 mM HEPES pH 8, 150 mM KCl buffer using a field of view 512 × 138 pixels, collecting 6000 frames with a collection time of 60 s. The focal position and imaging conditions were set using a 14 µL buffer droplet and data was collected by adding 2 µL of condensin sample, resulting in a final protein concentration of ~7 nM. All data were acquired with using the Refeyn AcquireMP software and analysed using the Refeyn DiscoverMP software. Masses were calibrated using the NativeMark unstained protein standard (LC0725, Thermo Scientific) to generate a calibration curve.

## ATPase activity assays

ATPase assays were performed based on the method described previously (Voulgaris and Gligoris, 2019) using the EnzCheck phosphate assay kit (Invitrogen). Condensin I complexes were used at a final concentration of 50 nM, while HAWK subunits were used at 100 nM and peptides were used at 50 µM. Basal ATPase assays were performed with final salt concentration of 100 mM KCl. DNA titrations were performed with 50 bp dsDNA annealed by heating to 98 degrees and slowly cooling in annealing buffer (10 mM Tris pH 7.5, 50 mM NaCl, 1 mM EDTA).

Plate was read with a POLARstar Omega plate reader (BMG).

## EMSA

The DNA sequence used for ATPase assays was used for gel shift assays, except a Cy5 label was added to the 5' of the reverse oligonucleotide (Reagents and Tools table). DNA was used at a final concentration of 50 nM, with indicated concentration of protein in buffer 20 mM HEPES pH 8, 150 mM KCl, 2.5 mM MgCl$_2$, 10% glycerol, 1 mM DTT. Sample was incubated on ice for 20 min before being resolved on a 2% agarose gel in 0.5× Tris borate (TB) buffer and scanned on a FUJIFILM FLA-5100 scanner.

## AlphaFold predictions

AlphaFold2 multimer predictions were performed using AlphaFold v2.3.1 using default parameters of the following Colab notebook with a Colab Pro account: https://colab.research.google.com/github/deepmind/alphafold/blob/main/notebooks/AlphaFold.ipynb.
https://colab.research.google.com/github/sokrypton/ColabFold/blob/main/AlphaFold2.ipynb.

Predictions of human condensin complex used residues NCAPG$_{1-912}$, NCAPH$_{1-101}$, NCAPD2$_{1307-1401}$ or KIF4A$_{1065-1232}$. Prediction of yeast condensin interaction used residue Ycg1$_{1-475}$ with full length Sgo1 or Lrs4.

Plots showing per residue pLDDT and per residue aligned error were generated using output PDB and JSON file using Python run in Jupyter notebooks. Figures of structural models were generated with ChimeraX, with regions of low prediction confidence hidden for clarity.

AlphaFold3 multimer predictions were performed using the AlphaFold3 webserver (Abramson et al, 2024) using the same regions of NCAPG, NCAPH, NCAPD2, Ycg1, Sgo1 and Lrs4. Predictions of long regions of KIF4A resulted in low confidence helical predictions, but stable, consistent predictions were attained with residues KIF4A$_{1136-1232}$ with NCAPG.

In all cases, AlphaFold3 predicted a comparable interface as AlphaFold2, with C-alpha RMSDs of 3.8 Å for KIF4A$_{1207-1232}$, 2.0 Å for NCAPH$_{38-79}$, 1.4 Å NCAPD2$_{1376-1395}$, 0.8 Å Lrs4 residues 317–331, and 0.7 Å for Sgo1$_{502-519}$. AlphaFold3 generally produced more confident pLDDT scores, predicting additional high-confidence contacts in Lrs4 and Sgo1 (Fig. EV3E,F,H,I), and between residues NCAPH$_{K40}$ with the acidic patch NCAPG and NCAPH$_{L44}$ with the hydrophobic pocket of NCAPG (Fig. EV2F,G). The interaction between KIF4A and NCAPG was predicted with lower confidence by AlphaFold3; the prediction was comparable for the basic patch, KIF4A$_{K1208-1211,R1212}$ and the hydrophobic pocket interaction, KIF4A$_{L1214}$, but was divergent for KIF4A$_{F1220,1221}$ (Fig. EV2J,K), despite these residues being shown to be required for KIF4A/NCAPG binding in cells (Poser et al, 2019). The results of AlphaFold3 could suggest the overlap between NCAPH and KIF4A may be more significant, and require additional regions of NCAPH.

### Fluorescence polarisation assays

Fluorescence polarisation binding assays were performed by mixing 100 nM of 5-FAM labelled peptide (Reagents and Tools Table) with indicated concentration of protein in FP buffer (20 mM HEPES pH 8, 150 mM KCl, 2.5 mM MgCl, 5% glycerol, 1 mM DTT) and incubated at room temperature for 15 min before reading the plate with a POLARstar Omega plate reader (BMG). The plate was read three times at 5 min intervals to ensure samples reached equilibrium and

three technical replicates were performed, setting up a new plate, with distinct protein aliquot in each case. Data was analysed and plotted in Graphpad Prism and Kd fit using the following:

$$FP = \left(\frac{FP_{max}}{2[Pep]}\right)\left(([C] + [Pep] + K_d) - \sqrt{([C] + [Pep] + K_d)^2 - 4.[C].[Pep]}\right)$$

Where [C] and [Pep] are the concentration, in µM, of condensin and 5-FAM labelled peptide respectively, $FP_{max}$ the maximum change in fluorescence polarisation in mP and $K_d$ is the equilibrium dissociation constant. Fit curves are plotted, with standard deviation.

### Fluorescence polarisation competition assays

FP competition assays were performed for NCAPD2 peptide, using a fixed concentration of 100 nM of F-NCAPD2-C$_{1376-1398}$ and 6 µM of NCAPG, in the presence or absence of 10 µM KIF4A WT$_{1206-1228}$ or KIF4A Mut$_{1206-1228}$ peptide. Similar assays were performed for NCAPH2 peptide, using a fixed concentration of 100 nM of F-N-NCAPH$_{55-77}$ and 4 µM CI NdelH, D2delC, in the presence or absence of 100 µM of KIF4A WT$_{1206-1228}$ or KIF4A Mut$_{1206-1228}$ peptide.

Background was determined by measuring 100 nM of F-NCAPD2-C$_{1376-1398}$ or F-N-NCAPH$_{55-77}$ in the presence of peptide buffer (20 mM HEPES pH 8), KIF4A WT$_{1206-1228}$ or KIF4A Mut$_{1206-1228}$ peptide, and subtracted from each measurement. Assay was performed with three technical repeats, setting up a new plate with distinct protein aliquot each time, reading the plate three times to ensure equilibrium. Data analysed and plotted using GraphPad Prism.

### Single-molecule assay DNA substrate preparation

The 48.5 kbp DNA substrate used in the single-molecule assay in this study was prepared as follows. Linear and double-stranded (48502 bp) Lambda Phage DNA with 12 bp single-stranded 5'-ends was purchased from New England Biolabs (N3011L). The complementary biotinylated primers were ligated with Taq DNA Ligase to achieve biotinylation on both ends of the Lambda DNA. The Lambda DNA biotinylated at both ends was purified from the 10× molar excess primers and the ligase by a custom-built Äkta column.

### Single-molecule flow cell preparation

The microscope slides (76 × 26 × 1 mm microscope slides Marienfeld 1000000) and coverslips (24 × 60 mm No 1.5 thickness 170 µm, borosilicate VWR Intl 631-0147) used in this study were functionalized as previously described (Chandradoss et al, 2014) with the minor following modifications. Microscope slides were laser-drilled to achieve 11 channels to attach inlet / outlet tubings and reused several times. Since an objective-type TIRF microscope was used the coverslips were also treated with 1 M of KOH solution and etched with acid Piranha solution of 5:1 sulphuric acid:hydrogen peroxide ratio prior to amino-silanization. The microscope slides and coverslips were silanized with a solution of 100 mL of anhydrous methanol, 5 mL of acetic acid and 10 mL of APTES ((3-aminopropyl)triethoxysilane) and PEGylated with 4 mg of m-PEG-SVA (MW 5000 Laysan Bio) and 0.1 mg Biotin-PEG-SVA (MW 5000 Laysan Bio) in 50 mM boric acid, 12.5 mM Sodium Hydroxide pH 8.5. The PEGylation was repeated four times for at least 4 h at 4 °C. The microscope slides, and coverslips were washed with MQ water and dried with a gentle flow of nitrogen between the

PEGylation steps. After the 4th PEGylation the microscope slides and coverslips were sealed and stored at −20 °C until the assembly of the flow cell. Prior to the flow cell assembly, a 5th PEGylation step using 50 mM MS(PEG)4 (Thermo Scientific) in 50 mM Boric Acid, 12.5 mM Sodium Hydroxide pH 8.5 was done overnight at 4 °C. The shorter (MW 333) PEG molecules were used in the last PEGylation step since they may be more effective in filling up possible holes in the PEG layer. After this step, the microscope slide and coverslip were washed with MQ water, dried with a gentle flow of nitrogen and used directly in the assembly of a flow cell.

Multi-channel flow cells were assembled using the functionalized microscope slides and coverslips with double-sided scotch tape as a spacer between the two to achieve 2 mm × 24 mm × 100 µm channels. The channels were sealed with Epoxy glue. The outlets were assembled using 1–200 µl Axgen pipette tips, tubings and 0.5 × 10 mm syringe needles glued together with Epoxy, attached to the laser-drilled holes of the microscope slide and glued in place with Epoxy. 1–200 µl Axygen pipette tips were used as inlet during experimenting.

### HiLo microscopy and data acquisition

The HiLo microscope used in this study was described previously (Pradhan et al, 2023).

### Single-molecule loop extrusion assays

Real-time imaging of the loop extrusion by condensin I was carried out in the aforementioned flow cells as follows. The channel was washed with T50 buffer (40 mM Tris-HCl pH 8.0, 50 mM NaCl, 0.2 mM EDTA), incubated with 1 µM Streptavidin in T50 buffer for 2 min and incubated with 5 mg/ml BSA in T50 buffer for 5 min to further reduce unspecific binding of proteins to the surface. The excess Streptavidin and BSA were washed thoroughly with T50 buffer. Lambda DNA biotinylated at both ends was introduced to the channel at a flow rate of 2 µl/min facilitating surface attachment through the streptavidin–biotin binding. The flow was adjusted and kept constant with a syringe pump attached to the outlet of the flow cell. Excess DNA was washed thoroughly with T50 buffer.

The surface attached DNA molecules were imaged in the imaging buffer (40 mM Tris-HCl pH 7.5, 2 mM Trolox, 30 mM D-glucose, 50 mM Potassium Glutamate (L-Glutamic acid mono-potassium salt monohydrate), 2.5 mM MgCl$_2$, 1 mg/ml BSA, 1 mM TCEP, 2.5 mM ATP, 100 nM Sytox Orange, 30 mg/ml Glucose Oxidase (15 U/ml), 20 mg/ml Catalase (1000 U/ml)). The imaging buffer supplied with used condensin I variant was introduced into the channel at a flow rate of 10 µl/min for 100 s and the flow was stopped afterward. For the activity assay comparing condensin I WT and condensin I ΔNCAPG 1 nM of protein, for the activity assay comparing condensin I WT and condensin I WT supplied with KIF4A peptide 0.5 nM of protein and 1000× of the KIF4A peptide, for the analysis of loop extrusion rate 0.75 nM–3 nM of protein (in the case of condensin I WT supplied with KIF4A peptide, correspondingly 1000× of KIF4A peptide) was used.

### Single-molecule data analysis

A custom-written Python code was used to analyse the fluorescence image series. (Pradhan et al, 2023) Python packages Napari (https://github.com/Napari/napari) and PyQtGraph (https://github.com/pyqtgraph/pyqtgraph) were used for the visualisation and evaluation of the data.

### Analysis of loop extrusion activity

Region of interests (ROIs) containing a single Lambda DNA molecule were chosen manually and annotated using the tools provided by the Python package Napari. For a single loop extrusion experiment, a field of contains ~100–200 DNA strands. Only the strands that were tethered at both ends from the beginning to the end of the data acquisition were considered for the analysis. Each of the DNA strands were manually analysed whether they form at least one loop during an acquisition time of 1000 s, including the initial 100 s of flow. The "fraction of looped DNA" (Fig. 4B,C) was defined as the ratio of the number of the DNA strands which form at least one loop to the total number of DNA strands. This quantity was taken as a measure for the activity of the protein. For the average fraction of looped DNA (Fig. 4B,C) the mean of multiple, independent experiments was taken. (For 1 nM condensin I WT 4, for 1 nM condensin I ΔNCAPG 10, for 0.5 nM condensin I WT 3, For 0.5 nM condensin I WT supplied with 0.5 μM KIF4A three independent experiments were performed and analysed.) The error bars in the Fig. 4 are given by the standard deviation of these datasets. Since the number of independent data points for each of the variants was not sufficient to assume normality of the distribution we applied the non-parametric Wilcoxon rank-sum test to determine the p values. The stats package from the Python module scipy was used for the statistical tests.

### Analysis of loop extrusion rate

ROIs with a DNA strand that formed a loop were cropped and saved in TIFF format for further analysis. For snapshots shown in the Fig. 4A,D, a median filter with a radius of 2 pixels was applied for smoothing the image. Furthermore, a white top-hat filter with radius 10 was applied for background subtraction using the "white_tophat" function from the package ndimage from the Python module scipy. The kymographs were built by summing the fluorescence intensity for 11 pixels along the line centred around the DNA axis from the median filtered images. Each vertical line in the kymograph corresponds to one frame of the image series.

For the determination of the loop extrusion rate, each line in the kymograph was split into three regions: "Up", "Loop" and "Down" (Fig. 4D). For this, first the loop region was determined as follows: Using the "find_peaks" function from the package signal from the Python module scipy local intensity maxima in each line of the kymograph was determined. The centre position of the DNA loop was chosen as the most intense peak within one line. The single peaks along the kymograph were connected using the "link" function from the Python package trackpy while manually supervising that the connected peaks correspond to the extruded loop. The region "Loop" was then assigned to the 9 pixels along the centre peak for each line of the kymograph. The regions "Up" and "Down" were assigned to the remaining upper and lower portion of the kymograph, respectively for each line. The amount of DNA in each of the regions was estimated by multiplying the ratio of fluorescence intensity of each region to the total fluorescence intensity with 48.5 kbp which corresponds to the total length of the used Lambda DNA strands. The DNA size in each region against time gives the kinetics of a loop formation. In Fig. 4F, the kinetics of an exemplary loop is plotted together with the smoothed data using a Savitzky–Golay second-order filter and 50 points window size. The loop extrusion rate in Fig. 4G was determined by fitting a linear function $f(x) = kx + c$ to the initial 5 s of the loop growth curve. The 5 s window of the fitting was chosen after the initiation of the loop where the loop size was increasing monotonously. Loop formation events were left out from the analysis of the loop extrusion rate if the signal was not clear enough which could be due to several reasons, e.g., other DNA strands in the way, the loop formation event is not very stable and the loop slides back within the initial 5 s window, the loop forms at the upper or lower edge of the kymograph making it not possible to determine the size of the size of the "Up" or "Down" portion, respectively, etc.

## Data availability

Raw biochemical assay and imaging data used in figures have been submitted as source data. Analysis scripts for loop extrusion data have been previously published (Pradhan et al, 2023). Any other original imaging data is available on request.

The source data of this paper are collected in the following database record: biostudies:S-SCDT-10_1038-S44318-024-00340-w.

## Peer review information

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

## Acknowledgements

The authors thank David Rueda, Paul Girvan and Anita Meier for assistance in collecting proof of principal single-molecule data, and Alessandro Vannini for human condensin I constructs. This study was funded by the Medical Research Council UKRI MC-A652-5PY00 (LA and EEC), the Max Planck Society (EK), European Research Council Starting Grant 101076914 (EK) and Deutsche Forschungsgemeinschaft (DFG, German Research Foundation)—SFB 1551 (DT and EK).

## Author contributions

**Erin E Cutts**: Conceptualisation; Data curation; Formal analysis; Investigation; Visualisation; Methodology; Writing—original draft; Writing—review and editing. **Damla Tetiker**: Data curation; Formal analysis; Investigation; Visualisation; Writing—review and editing. **Eugene Kim**: Resources; Supervision; Funding acquisition; Writing—review and editing. **Luis Aragon**: Resources; Supervision; Funding acquisition; Writing—review and editing.

Source data underlying figure panels in this paper may have individual authorship assigned. Where available, figure panel/source data authorship is listed in the following database record: biostudies:S-SCDT-10_1038-S44318-024-00340-w.

## Disclosure and competing interests statement

The authors declare no competing interests.

# Expanded View Figures

**Figure EV1.  Condensin I mutant complex quality control.**

(**A**) Comparison of two different batches of recombinant CIWT pentamer, and CIΔG and CIΔD2 tetramers. Trend of tetrameric complexes being significantly more active than pentameric condensin I is consistent. Data from three technical replicates. Error bars represent s.e.m. *P* values indicated are from unpaired, two-tailed t test with Welch's correction. (**B**) Mass photometry data of condensin I tetramers, CIΔG, with and without NCAPG being added to reconstitute the pentamer. (**C**) Mass photometry data of condensin I tetramers, CIΔD2, with and without NCAPD2 being added to reconstitute the pentamer. (**D–F**) Mass photometry data of condensin I pentamers, CID2ΔC, CIHΔN and CIHΔN,D2ΔC, respectively, consistent with pentameric mass and confirming they are the major species present. Source data are available online for this figure.

**A**

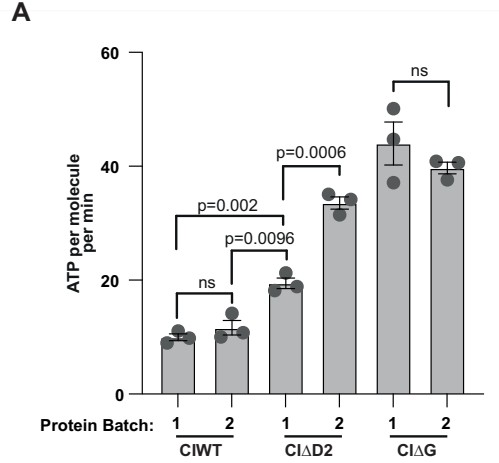

**B**

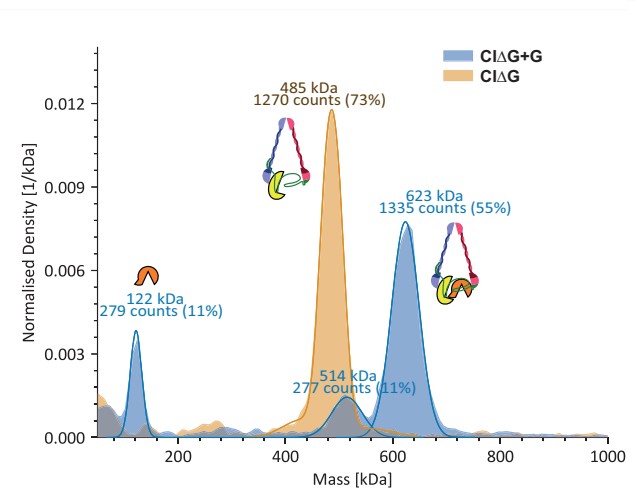

**C**

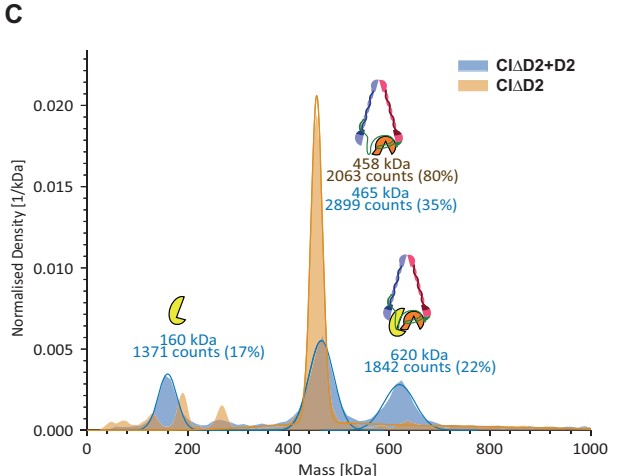

**D**

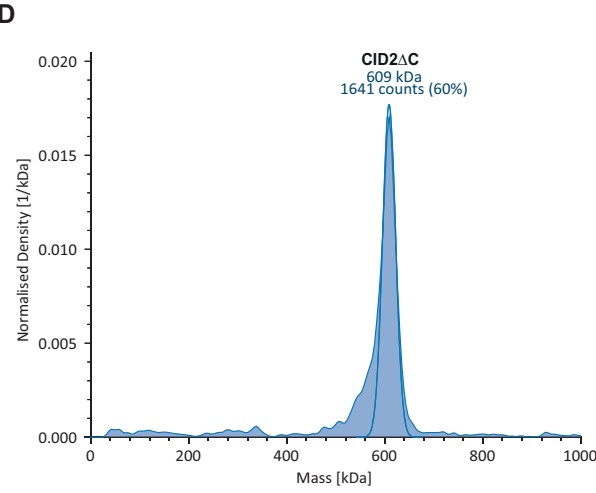

**E**

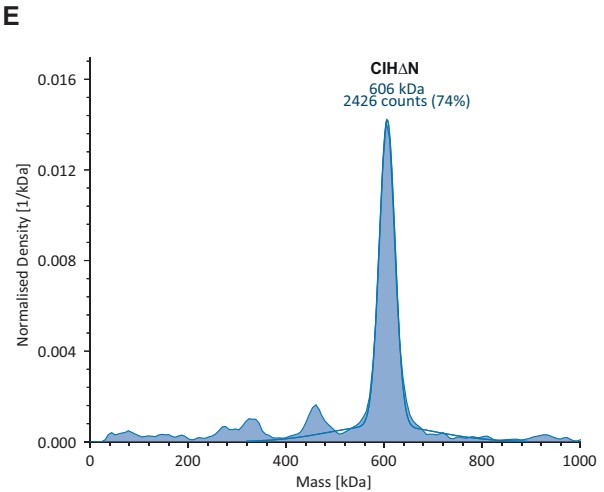

**F**

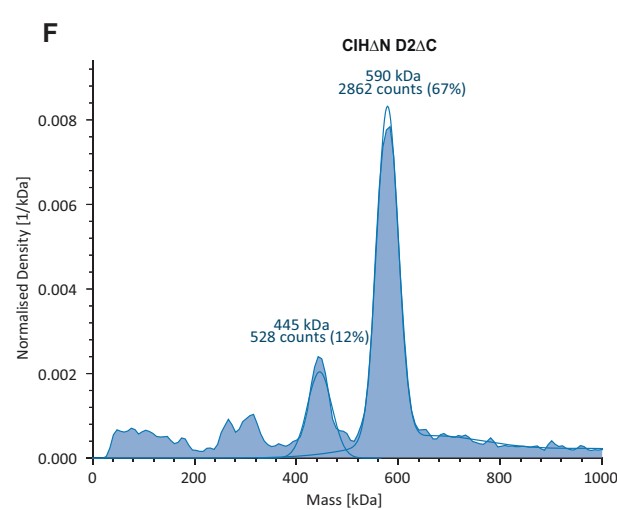

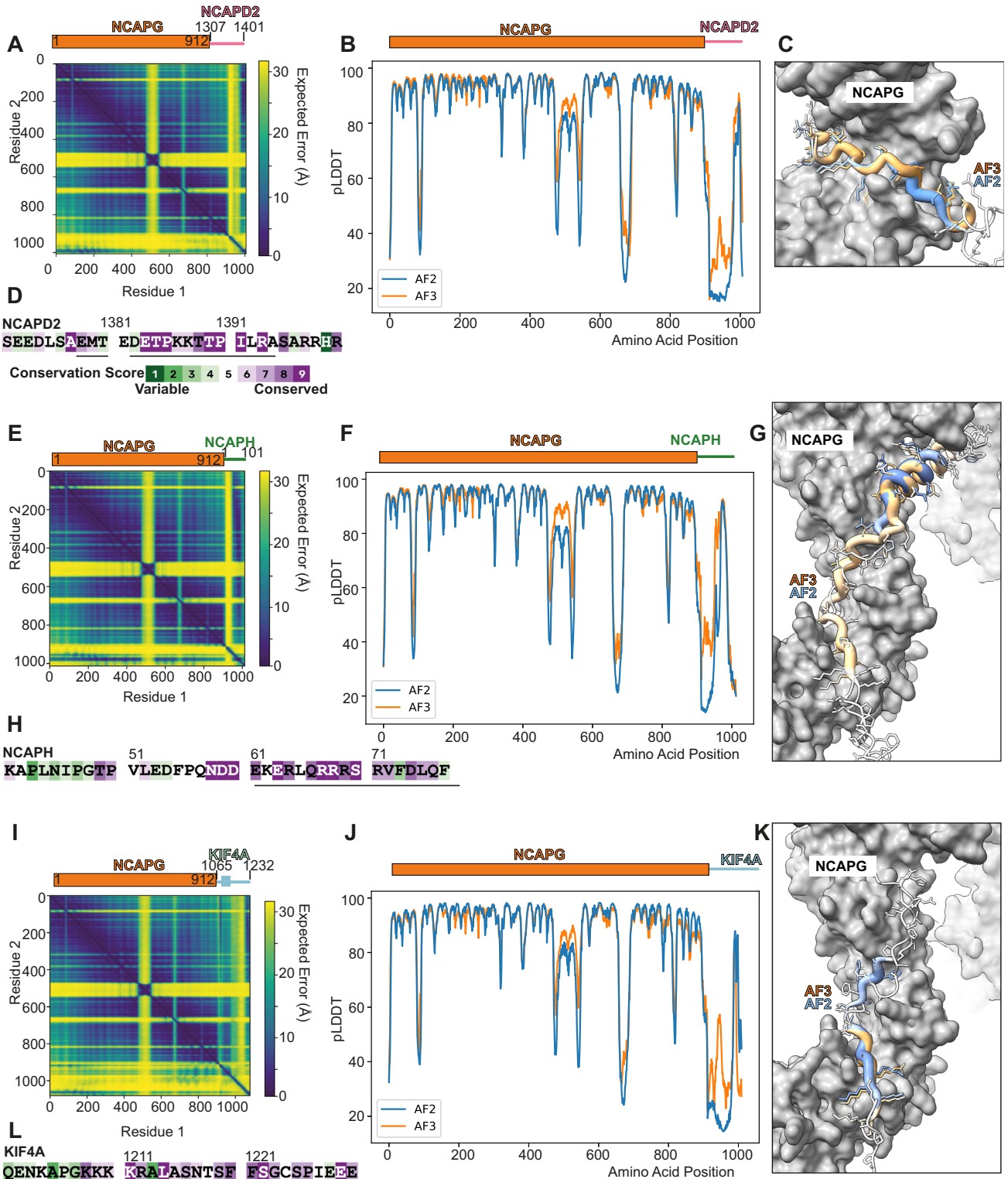

**Figure EV2. NCAPG SLiM interaction AlphaFold2 and 3 confidence.**

(A) AlphaFold2 predicted alignment error plot for interaction between NCAPG and NCAPD2$_{1307-1401}$. (B) Predicted local distance difference test (pLDDT) score per residue of AlphaFold2 and 3 (AF2 and AF3, respectively) predictions of NCAPG and NCAPD2$_{1307-1401}$. (C) Comparison of AF2 (blue) and AF3 (orange) predictions of NCAPG and NCAPD2$_{1307-1401}$. AlphaFold pLDDT score is represented in thickness of NCAPD2 backbone and colour intensity. (D) SLiM region with conservation colouring from ConSurf (Ashkenazy et al, 2016). (E–H) Equivalent data to (A–D) for prediction of NCAPG with NCAPH$_{1-101}$. (I) AlphaFold2 predicted alignment error plot for interaction between NCAPG and KIF4A$_{1065-1401}$. (J–L) Equivalent data to (B–D) for prediction of NCAPG with KIF4A$_{1136-1232}$.

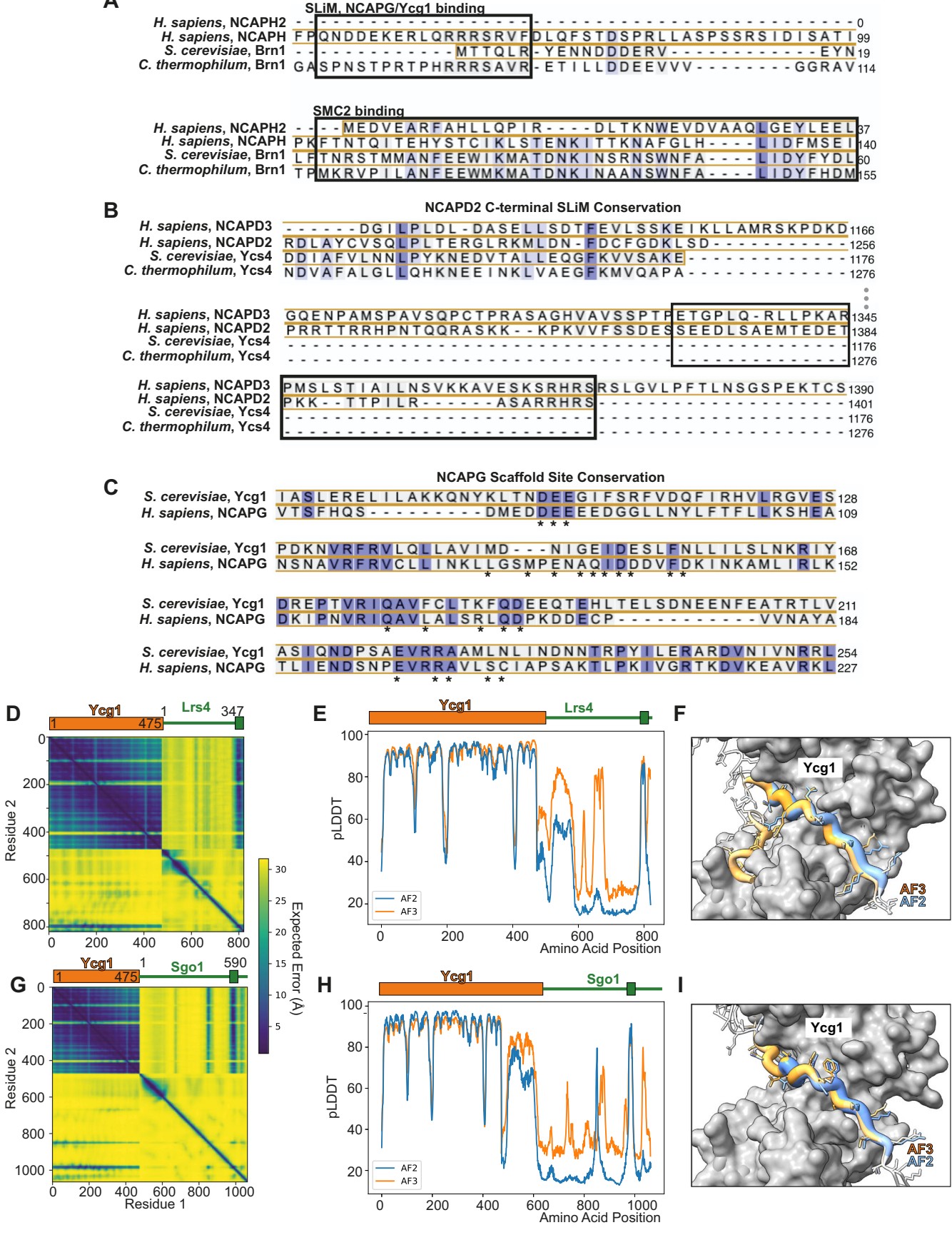

◀ **Figure EV3.  Condensin disordered sequence conservation.**

(A) Alignment of N-terminal region of NCAPH/H2 and Brn1, in *H. sapiens, S. cerevisiae* and *C. thermophilium*, with NCAPG/Ycg1 and SMC2 binding regions indicated. (B) Alignment of C-terminal region of NCAPD2/3 and Ycs4 in *H. sapiens, S. cerevisiae* and *C. thermophilium*. (C) Alignment of SLiM docking region in NCAPG and Ycg1, from *H. sapiens and S. cerevisiae*, (*) indicates residues with in 4 Å of predicted bound SLiM. (D) AlphaFold2 predicted alignment error of Ycg1$_{1-475}$ and Lrs4. (E) Predicted local distance difference test (pLDDT) score per residue of AlphaFold2 and 3 (AF2 and AF3, respectively) predictions of Ycg1$_{1-475}$ and Lrs4. (F) Comparison of AF2 (blue) and AF3 (orange) predictions of Ycg1$_{1-475}$ and Lrs4. AlphaFold pLDDT score is represented in thickness of Lrs4 backbone and colour intensity. (G–I) Equivalent data as (D–F) for Ycg1$_{1-475}$ and Sgo1. Sequence data sourced from Uniprot (Bateman et al, 2023).

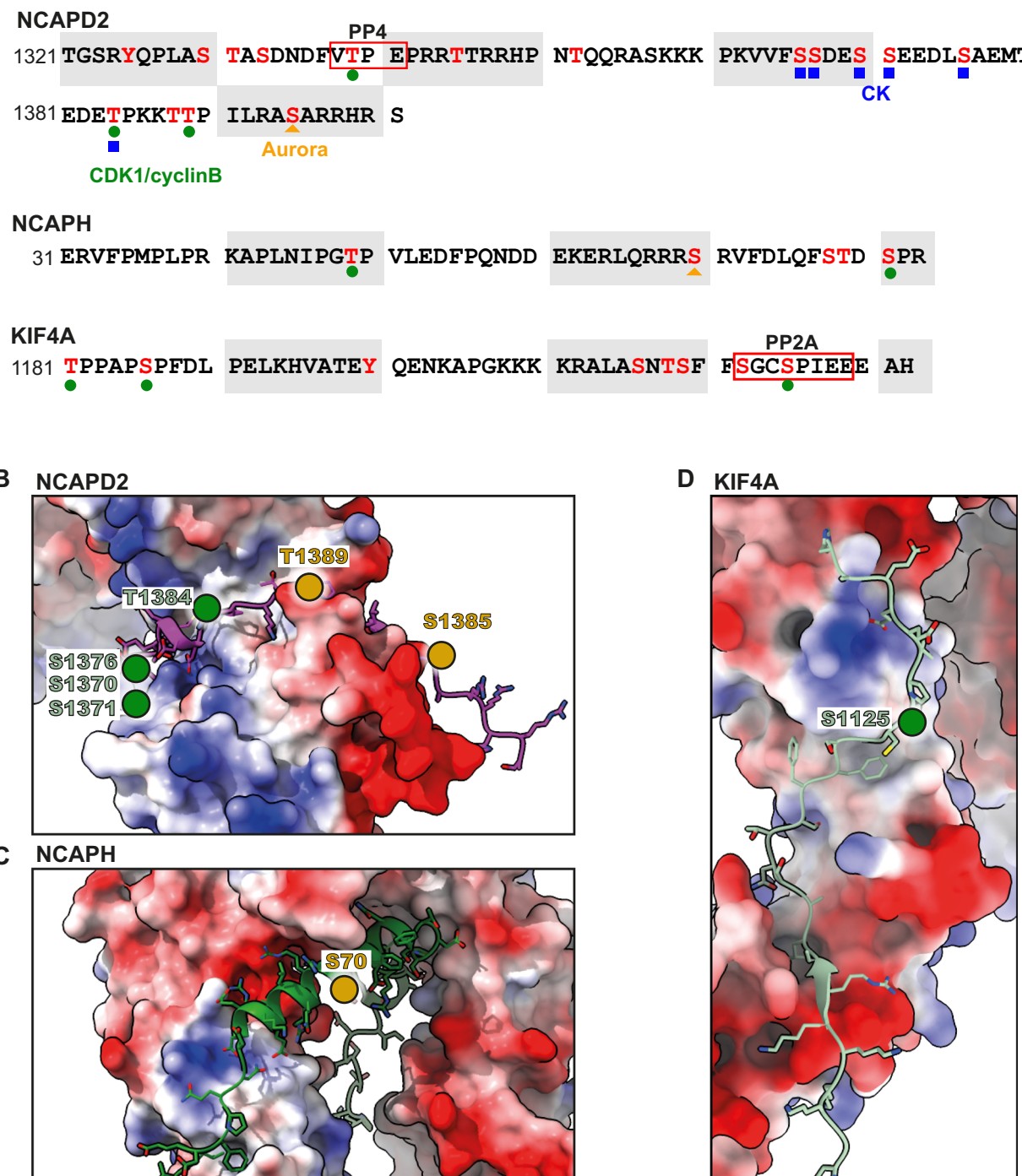

**A**

**NCAPD2**

1321 TGSR**Y**QPLA**S** TA**S**DNDF**VTP** E**PRRT**TRRHP NT**QQ**RASKKK PKVVF**SS**DE**S** **S**EEDL**S**AEMT

1381 EDE**T**PKK**TT**P ILRA**S**ARRHR S

CDK1/cyclinB

Aurora

CK

PP4

**NCAPH**

31 ERVFPMPLPR KAPLNIPG**TP** VLEDFPQNDD EKERLQRRR**S** RVFDLQF**ST**D **S**PR

**KIF4A**

1181 **T**PPAP**S**PFDL PELKHVATE**Y** QENKAPGKKK KRALA**S**NT**S**F F**SGCS**PIEEE AH

PP2A

**B  NCAPD2**

T1389
T1384
S1385
S1376
S1370
S1371

**C  NCAPH**

S70

**D  KIF4A**

S1125

Charge
-                    +

◀  **Figure EV4.  Phosphorylation and phosphatase sites in SLiMs.**

(**A**) Theoretical sites of phosphorylation by Aurora (orange triangle), CDK1/cyclin B (green circle) and casein (blue square) kinases in SLiM of NCAPD2, NCAPH and KIF4A. Red rectangles indicate phosphatase binding sites. (**B–D**) The location of phosphorylation sites indicated in (**A**) in NCAPD2, NCAPH and KIF4A, respectively. Based off of electrostatic surface charge, the effect of the phosphorylation is coloured either green, for potentially enhancing the interaction or orange, for resulting in a potential charge clash.

