## [Peer Review File · The EMBO Journal]

Molecular Mechanism of Condensin I Activation by KIF4A

Luís Aragón, Erin Cutts, Damla Tetiker, and Eugene Kim

Corresponding author(s): *Luís Aragón (Luis.aragon@lms.mrc.ac.uk)* , *Erin Cutts (e.cutts@sheffield.ac.uk)*

Review Timeline:

Submission Date:	22nd May 24
Editorial Decision:	9th Jul 24
Revision Received:	9th Oct 24
Editorial Decision:	23rd Nov 24
Revision Received:	4th Dec 24
Accepted:	6th Dec 24

Editor: Hartmut Vodermaier

Transaction Report:

Dr. Luís Aragón
MRC Laboratory of Medical Sciences
Du Cane Road
London, London W12 0HS
United Kingdom

9th Jul 2024

Re: EMBOJ-2024-117957-T
Molecular Mechanism of Condensin I Activation by KIF4A

Dear Drs. Aragon and Cutts,

Thank you for submitting your manuscript on condensin activation by KIF4A for our consideration. With some delay due to extra time needed for back-to-back reviewing, we have now received the below-copied reports from three expert referees. As you will see, while referees 1 and 3 are generally positive about the study, referee 2 remains still more critical at this stage. Given that many of their concerns appear to be related to presentational issues and insufficient clarity of experimental conditions, I would nevertheless like to give you an opportunity to respond to the reviewers' comments by way of a revised manuscript. However, please note that it is our policy to consider only a single round of major revision, making it important to satisfactorily clarify all main points at the time of resubmission. In this light, it should be beneficial if you contacted me with a tentative point-by-point response already during the early stages of your revision, so that we could discuss how the key issues would best be addressed. Should this require more than our default three-month revision period, we would be happy to offer an extension, during which your study would at EMBO Press still remain protected from scooping by any competing works appearing in the meantime.

Further information on preparing, formatting and uploading a revised manuscript can be found below and in our Guide to Authors. Thank you again for the opportunity to consider this work for The EMBO Journal, and I look forward to hearing from you in due time.

With best regards,

Hartmut

4) Each main and each Expanded View (EV) figure should be uploaded as individual production-quality files (preferably in .eps, .tif, .jpg formats). For suggestions on figure preparation/layout, please refer to our Figure Preparation Guidelines:

- 5) Point-by-point response letters should include the original referee comments in full together with your detailed responses to them (and to specific editor requests if applicable), and also be uploaded as editable (e.g., .docx) text files.
- 6) Please complete our Author Checklist, and make sure that information entered into the checklist is also reflected in the manuscript; the checklist will be available to readers as part of the Review Process File. A download link is found at the top of our Guide to Authors: embopress.org/page/journal/14602075/authorguide
- 7) All authors listed as (co-)corresponding need to deposit, in their respective author profiles in our submission system, a unique ORCID identifier linked to their name. Please see our Guide to Authors for detailed instructions.
- 8) Please note that supplementary information at EMBO Press has been superseded by the 'Expanded View' for inclusion of additional figures, tables, movies or datasets; with up to five EV Figures being typeset and directly accessible in the HTML version of the article. For details and guidance, please refer to: embopress.org/page/journal/14602075/authorguide#expandedview
- 9) To facilitate reproducibility and cross-laboratory adoption of methodologies, please structure the Materials & Methods section as outlined in our guide to authors, including a completed Reagents and Tools Table that can be downloaded from our author guidelines as well (<https://www.embopress.org/page/journal/14602075/authorguide#structuredmethods>).
- 10) Digital image enhancement is acceptable practice, as long as it accurately represents the original data and conforms to community standards. If a figure has been subjected to significant electronic manipulation, this must be clearly noted in the figure legend and/or the 'Materials and Methods' section. The editors reserve the right to request original versions of figures and the original images that were used to assemble the figure. Finally, we generally encourage uploading of numerical as well as gel/blot image source data; for details see: embopress.org/page/journal/14602075/authorguide#sourcedata

At EMBO Press, we ask authors to provide source data for the main manuscript figures. Our source data coordinator will contact you to discuss which figure panels we would need source data for and will also provide you with helpful tips on how to upload and organize the files.

In the interest of ensuring the conceptual advance provided by the work, we recommend submitting a revision within 3 months (7th Oct 2024). Please discuss the revision progress ahead of this time with the editor if you require more time to complete the revisions. Use the link below to submit your revision:

Link Not Available

Referee #1:

Condensin complexes play an important role in chromatin compaction in mitosis. It has been known that the chromosomal association of Condensin I, which consists of SMC2/4, NCAPH, NCAPD2 and NCAPG, depends on the chromokinesin KIF4A. This study sought to dissect the underlying molecular mechanism. The authors find that KIF4A utilizes a conserved disordered SLiM in its C-terminus to bind the HAWK subunit NCAPG of Condensin I. This interaction seems to compete with the auto-inhibitory interaction of NCAPG with NCAPH and NCAPG2. Moreover, a KIF4A peptide containing the SLiM is able to promote the ATPase and DNA loop extrusion activity of Condensin. The authors provide further evidence showing that SLiM-mediated binding to NCAPG is a conserved mechanism of regulation in SMC complexes. Overall, this is a nice study which elegantly addresses an important question in the field. The conclusions are largely supported by the data.

Major points:

1. In line 113, the authors describe that "The hyperactivity of tetrameric complexes was rescued to near pentameric levels when an excess of recombinant NCAPG or NCAPD2 were added", and that "Adding excess NCAPG or NCAPD2 to pentameric CI had no effect, demonstrating that super stoichiometric HAWK subunits did not further enhance repression". However, I have a little trouble to understand these description based on the data shown in Figure 1b.
2. Authors used AlphaFold2 to predict several protein-protein interaction models. It would be nice if the authors could test whether AlphaFold3-predicted models are the same as those predicted by AlphaFold2.
3. Based on the results obtained from fluorescence polarisation assays (Figure 3D), the authors suggest that "KIF4A competes

with NCAPD2 to bind NCAPG", It would be nice if the authors could perform biochemical protein-protein interaction assays to support this conclusion.

Minor points:

1. The writing of manuscript needs to be improved.
2. It would be nice if the authors could discuss the evolutionary conservation of the Sgo1-Ycg1 interaction.
3. In lines 284-286, the authors mention that "We have shown that NCAPG acts as a scaffold, binding multiple SLiMs. The equivalent subunits in condensin II and cohesin, NCAPG2 and STAG1/2, respectively, also act as a scaffold for SLiM interactions^{21,33,34}". A recent study reported that the inner kinetochore protein CENP-U utilizes a F-D-F motif to bind the composite interface between SA2 and Scc1 (PMID: 38714893) could also be cited.

Referee #2:

In the manuscript 'Molecular Mechanism of Condensin I Activation' Cutts et al. study condensin I regulation. The authors discuss the importance of the NCAPG subunit of condensin I, which is shown to bind two other subunits - NCAPH and NCAPD2, and to a previously identified condensin I regulator - KIF4A. The manuscript looks into the contribution of these regulators to condensin's ATPase and loop extrusion activity, and investigates the molecular interplay between these interactions.

The regulation of condensin I action via the N-terminus of NCAPH, and of condensin II via the C-terminus of NCAPD3, has been recently discussed in Tane et al. 2022 and Yoshida et al. 2022 respectively. Building on these already existing data, the manuscript by Cutts et al. provides molecular insight into how concerted action of NCAPH, NCAPD2 and KIF4A regulates condensin I activity.

The data overall is interesting and will be of value to the field. The presentation of the data however is not as good as it could have been. The manuscript is written in a rather confusing manner. Some conclusions are too strong and are not supported by the experiments. There are also multiple errors throughout the manuscript, both in the text and in the figures. This unfortunately makes what could have been a nice story appear rather confusing, and on some occasions this makes me doubt the validity of the data. So in my view quite some work would be needed before this manuscript is suited for publication in EMBO Journal.

Major issues:

I have a number of major concerns regarding figure 1b:

- First, the gel appears to be a combination of multiple cut out gels, which should be indicated properly. Otherwise, the figure creates an improper assumption that all the tested conditions belong to one experiment, which presumably is not the case. There should be visible spaces between the gels and/or explanation in the text or figure legend.
- Related to this point, the condition with all three pluses is present three times (column 1, 3 and 8). I assume this is done because these are in fact separate gels from potentially separate preps? These three columns have quite different ATP/molecule/min values. Some variability between experiments is understandable, but when the control between all three experiments is so different, a fair comparison of other samples between the different samples, cannot be made. For example, when comparing column 1 with column 2, authors state that this is a meaningful difference. But then, column 8, which should be a control, is pretty much the same value as column 2, which is where we should see a significant difference according to the authors. Comparing bar 1 with bar 7 is also not fair in this case.
- Labeling of the bands requires improvement as it is not clear which band belongs to which protein. In the first 4 lanes there is an extra band (or even two). Is this a degradation product? Also, due to the fact that there are multiple gels that have been glued together in this figure, it looks like the bands are not aligned properly, which makes for a very confusing unit. For example, in column 5 we should only see a band corresponding to NCAPD2. But this band runs at the height of SMC4 in other columns, for example 2.
- And more importantly: in column 7, in which the NCAPG subunit should not present, the band is still there. Is this a mistake or how can this be explained?

All the data presented in figure 1b are potentially exciting, but until all the points mentioned above have been addressed, the figure remains confusing. These various points also make me doubt the validity of the data. If there simply is lots of experimental variation between the different wild type preps, then an equal number of preps is also needed for all the mutant preps. And it would be essential to plot the relative spread of the results both between preps and within preps, depicting the individual data points for all. And then please load all on one gel, with the correct labels for each sample.

Further points on figure 1:

- In figures 1b and 1c, the authors explain that ATP activity assays are performed. It is clear that the difference between these experiments is the presence of DNA. But these two panels also show the measurements in a different way, which would be needed to have explained to the reader.
- With regards to figure 1c, I find it confusing that the CIΔG and CIΔD2 conditions were not saturated with DNA to the same extent as the CI condition. What would the fold stimulation of ATPase rate look like for CIΔG and CIΔD2 at 13uM?
- In extended data figure 1b, there is a peak at about 144kDa corresponding to NCAPD2 subunit alone, a peak around 414kDa of Condensin I missing NCAPD2 subunit, and also a small peak of 544kDa of the reconstituted pentamer. First, the reconstituted

pentamer is a very small fraction, only 7%, which is a bit worrying. Secondly, what does the peak at 255kDa correspond to? It represents the biggest fraction the sample.

- In figure 1d, for specifically the C1ΔD2 samples, the bands shift all the way up into the slots. Why is that? Does this ΔD2 setting cause the complex to multimerise? Please comment.

Figure 2:

- In figure 2a in the C1 D2ΔC, and also in the C1 D2ΔC QL sample, the C-terminus of NCAPD2 is removed. That is a deletion of 1305 to 1401 amino acid. Because of that the protein can be expected to run lower on the gel, but would it be so much lower that it overlaps with the SMC2-corresponding band? Is it possible to run the gel longer or on a different percentage gel to better separate out the proteins? Now it looks as if C1 D2ΔC didn't express the NCAPD2 subunit at all. Also, the abbreviation of C1HΔN QL and C1ΔD2 QL is different than in the figure legends. Ideally this would be consistent, also with the Q-loop mutants in figure 1b.

- In figures 1b and 2a, it would be helpful to depict individual data points as well. Are the values represented by graph bars in figure 1b a representation of multiple replicates? And what then constitutes a replicate?

- Figure 2d shows that the NCAPG subunit alone binds the NCAPD2 peptide very efficiently. Condensin I pentamer binds the peptide much less well, which the authors speculate may be due to the fact that the NCAPD2 already present within the complex subunit is occupying NCAPG, and that the fluorescently labeled peptide cannot compete with it. How do authors then explain that in figure 2g with the NCAPH peptide this does not seem to be the case? Condensin I pentamer appears to bind NCAPH peptide even more efficiently than the NCAPG subunit alone. NCAPH peptide can then compete for binding to NCAPG with the NCAPH present within the pentamer? An note that a line connecting the data points for C1ΔG is missing in figure 2g.

- For figures such as 2d and 2g, please (also) plot the data with a log-scale axis, as this should allow for an S-shape of the curve. This benefits the K_d calculation/visualization.

Figure 3:

• In figure 3d, the authors use NCAPG protein with mutations D136K, D137K and D141K. These mutations were introduced in the acidic patch that was predicted to bind KIF4A by AlphaFold. This construct is used to show the abrogated binding between NCAPG and the KIF4A peptide. Could this mutant be used in binding assays with the NCAPD2 peptide? This is supposedly where NCAPD2 is predicted to bind NCAPG according to AlphaFold structures. This would further strengthen the data from figure 2d and provide an additional layer of evidence for the same binding spot on NCAPG for NCAPD2 and KIF4A.

• Figure 3e: The data shows that deletion of C-terminus of NCAPD2 subunit results in higher ATPase activity. The authors propose that this may be due to the fact that the NCAPG is then unoccupied, which allows for KIF4A to bind and stimulate ATPase activity of the complex. If it was simply about competition between NCAPD2 and KIF4A, why is there a higher ATPase activity in D2delC also without addition of KIF4A? This difference is also present in figure 2a in C1 D2ΔC sample. This implies that it cannot be simply about occupying the spot where the activator is binding. I believe the model can still hold true because the ATPase activation is even higher upon addition of KIF4A peptide, but it cannot be the only explanation and this should be elaborated on in the text.

• Also figure 3e: I would like to ask for clarification what is plotted on the y axis. Is it the same scale as in figure 1b? The wording is different and not properly explained in the text or the figure legend. It would also be helpful to keep the labeling coherent, so 'D2delC' could be D2ΔC, like used previously.

Figure 5:

- The authors base their hypothesis on 'existing models' in which the loop extrusion initiation requires DNA threaded through a compartment formed by the kleisin (discussed in lines 276-278). No literature is referenced to support these 'existing models'. This referee is not aware of any such model. Are the authors maybe referring to Lee et al, NSMB 2020? But then that paper does not include loop extrusion experiments. Or perhaps the authors meant to refer to other papers? Either way, please make sure to cite the appropriate work.

- The drawing of the model could be significantly improved, for example in panel b it is not clear that NCAPD2 and NCAPH are contacting NCAPG. There are white blocks covering the NCAPG subunit. The same can be said about the c panel and KIF4A binding. It would be helpful to label the subunits of condensin I in the figures, as well as KIF4A.

At the end of the paragraph 'Motif binding is conserved across species' the authors discuss data available in literature to strengthen their point about the importance of certain regions in Sgo1 and Lrs4. It is not clear that the data discussed here is not part of this manuscript, I would disclose this better. Also the data is not explained well at all. What is a Q325stop mutation? What is smc2-157 mutation and why is it relevant? This needs to be properly explained to the reader.

In the abstract, line 27, authors say that KIF4A directly competes with N-terminus of NCAPH and C-terminus of NCAPD2. The competition between KIF4A and NCAPD2 is shown in figure 3d, in which KIF4A peptide binds with higher affinity to Condensin I lacking C-terminus of NCAPD2. However, the competition between KIF4A and N-terminus of NCAPH was never shown. The same wording is used in the introduction "...we show that competition between KIF4A and NCAPD2/H binding results in condensin I activation" as well as in the title of a paragraph starting in line 213. I am afraid this was not shown in the manuscript. Appropriate experiments must be performed to draw these conclusions, or the wording in the manuscript needs to be adjusted.

Minor issues:

- Affiliation 2 and 3 are identical
- The authors may wish to update the information in the introduction (line 88) and in figure 7, regarding a newly identified NCAPG2 binder and condensin II regulator, M18BP1.
- The sentence starting in line 191: "The interface between the NCAPD2 SLiM and NCAPG..." is very difficult to read. It is not clear whether the mentioned set of residues belongs to NCAPD2 or NCAPG. Please clarify.
- Similarly, the sentence in lines 223-227 is rather difficult to understand.
- Line 223 contains a typo in the word 'mediated'.
- In figure 2c, the amino acids of NCAPD2 are mislabeled. The two glutamic acids (E) should not be at positions 1382 and 1384 but 1381 and 1383. The same goes for the two arginines (R). Instead of 1398 and 1399 they should be at 1397 and 1398.
- In figure 2e, the amino acid D63 is mislabeled. At position 63 there is a glutamic acid (E).
- In figure 2h, the amino acids are mislabeled. It should be F1220 and F122.
- Is the color-coding in figure 3g not correct? Should the 'down-size' be red and 'up-size' blue? If so, then please have this corrected, or else please better explain?

Referee #3:

The condensin complexes compact mitotic chromosomes through loop extrusion. Whether and how the loop extrusion activities of condensin are regulated are not well understood. Using in vitro biochemical and single-molecule biophysical experiments, Cutts et al. show that the ATPase activity and loop extrusion of human condensin I can be activated by a small motif in the kinesin KIF4A. They also show that yeast condensin interactors, such as Lrs4 and Sgo1, contain a similar motif, suggesting that this motif is conserved.

Although the interaction between condensin I and KIF4A has been reported previously, the biochemical function of this interaction is unknown. The current study provides strong evidence to indicate that KIF4A directly regulates the biochemical activity of condensin I and suggests a potential mechanism. As such, it will be of great interest to scientists in the chromosome biology field and should be published.

I have the following suggestions that the authors may wish to address prior to publication.

Major points

- (1) I found it surprising that condensin I lacking NCAPG was still capable of loop extrusion. Is this consistent with data in human cells? Specifically, do cells lacking NCAPG exhibit partial chromosome condensation? In addition, how does the current models of loop extrusion by condensin explain this finding?
- (2) In Fig. 2g, the NCAPG-binding motif of NCAPH binds to free NCAPG alone and the intact condensin I with similar affinity. This suggests that this motif of NCAPH does not bind to NCAPG with sufficient occupancy in intact condensin in the ground state. The authors should discuss this discrepancy and its implications for their proposed autoinhibition model.
- (3) The model presented in Fig. 5 is speculative and adds little to the manuscript.
- (4) Fig. 6 and Fig. 7 can be combined. Better still, they should test whether Sgo1 or Lrs4 binding stimulates the loop extrusion and/or ATPase activity of the yeast condensin.

We thank the reviewers for their feedback on our manuscript, we have thoroughly addressed these concerns and included additional supporting data and discussion. We thank them for their time and are pleased that they found our work to be of interest to the field. Our point-by-point responses are below.

Referee #1:

Condensin complexes play an important role in chromatin compaction in mitosis. It has been known that the chromosomal association of Condensin I, which consists of SMC2/4, NCAPH, NCAPD2 and NCAPG, depends on the chromokinesin KIF4A. This study sought to dissect the underlying molecular mechanism. The authors find that KIF4A utilizes a conserved disordered SLiM in its C-terminus to bind the HAWK subunit NCAPG of Condensin I. This interaction seems to compete with the auto-inhibitory interaction of NCAPG with NCAPH and NCAPG2. Moreover, a KIF4A peptide containing the SLiM is able to promote the ATPase and DNA loop extrusion activity of Condensin. The authors provide further evidence showing that SLiM-mediated binding to NCAPG is a conserved mechanism of regulation in SMC complexes. Overall, this is a nice study which elegantly addresses an important question in the field. The conclusions are largely supported by the data.

Major points:

1. In line 113, the authors describe that "The hyperactivity of tetrameric complexes was rescued to near pentameric levels when an excess of recombinant NCAPG or NCAPD2 were added", and that "Adding excess NCAPG or NCAPD2 to pentameric CI had no effect, demonstrating that super stoichiometric HAWK subunits did not further enhance repression". However, I have a little trouble to understand these descriptions based on the data shown in Figure 1b.

We believe that the figure's labelling scheme has caused some confusion (as reviewer 2 also questions this Figure; see below). Previously, we use a table that indicates which subunits are present. We have revised this to specify which complexes, sub-complexes, or individual subunits were present, and added lane numbers both to the figure and to the in-text references for additional clarity (lines 117-133, for Fig 1).

The confusion may also stem from a lack of clarity regarding the differences between protein complexes that are co-expressed versus those that are reconstituted from individual subunits or sub-complexes. We have included an explanation to clarify these distinctions (lines 122-124).

2. Authors used AlphaFold2 to predict several protein-protein interaction models. It would be nice if the authors could test whether AlphaFold3-predicted models are the same as those predicted by AlphaFold2.

Indeed, AlphaFold3 predicts the same interface, with C-alpha RMSDs of the overlaid region between 0 and 4 Angstroms.

We have added overlays of pLDDT confidence plots and structural predictions in Fig. EV2 and 3. However, AlphaFold2 predictions seem more stringent. AlphaFold3 tends to 'hallucinate' structures in disordered regions due to its diffusive minimization, resulting in higher background confidence and potential false structural features. We have mentioned AlphaFold3 predictions in the main text (line 208 and

239) and have provided detailed comparisons between AlphaFold2 and AlphaFold3 predictions in the Materials and Methods section (from line 897). However, we feel it is better to retain the main text figures with AlphaFold2 predictions, as it was these predictions that we had based our experiments on.

3. Based on the results obtained from fluorescence polarisation assays (Figure 3D), the authors suggest that "KIF4A competes with NCAPD2 to bind NCAPG", It would be nice if the authors could perform biochemical protein-protein interaction assays to support this conclusion.

We have added a fluorescence polarisation competition assay (Fig. 3E). This assay uses a fixed concentration of NCAPG and 5-FAM labelled NCAPD2 peptide, resulting in a high FP signal in the absence of KIF4A. The FP signal is significantly reduced in the presence of a KIF4A peptide while is unchanged in the presence of a mutant KIF4a peptide. Demonstrating that KIF4A competes and displaces the 5-FAM NCAPD2 peptide.

We have also added FP binding data showing that mutation of the NCAPG acidic patch, which does not bind KIF4A (Fig. 3D), does not bind NCAPD2 (Fig. 2D), thus providing further evidence that KIF4A and NCAPD2 bind the same site on NCAPG.

Collectively these two pieces of evidence demonstrate that NCAPD2 and KIF4A peptides bind the same site and compete.

Minor points:

1. The writing of manuscript needs to be improved.

We have endeavoured to improve the writing via receiving comments from peers prior to resubmission.

2. It would be nice if the authors could discuss the evolutionary conservation of the Sgo1-Ycg1 interaction.

We have stated that the residues that make the Sgo1 binding pocket on Ycg1 are conserved in human NCAPG (Line 331). We have also included citation of the Marston lab's recent preprint (<https://doi.org/10.1101/2024.03.27.586992>) in the discussion, but as they state "*Although Sgo1 CR1 is conserved only among yeasts closely related to S. cerevisiae*", the interaction between Sgo1 and Ycg1 seems poorly conserved (from line 438).

3. In lines 284-286, the authors mention that "We have shown that NCAPG acts as a scaffold, binding multiple SLiMs. The equivalent subunits in condensin II and cohesin, NCAPG2 and STAG1/2, respectively, also act as a scaffold for SLiM interactions^{21,33,34}". A recent study reported that the inner kinetochore protein CENP-U utilizes a F-D-F motif to bind the composite interface between SA2 and Scc1 (PMID: 38714893) could also be cited.

We have added this reference, which indeed supports our general model in Fig. 6F.

Referee #2:

In the manuscript 'Molecular Mechanism of Condensin I Activation' Cutts et al. study

condensin I regulation. The authors discuss the importance of the NCAPG subunit of condensin I, which is shown to bind two other subunits - NCAPH and NCAPD2, and to a previously identified condensin I regulator - KIF4A. The manuscript looks into the contribution of these regulators to condensin's ATPase and loop extrusion activity, and investigates the molecular interplay between these interactions.

The regulation of condensin I action via the N-terminus of NCAPH, and of condensin II via the C-terminus of NCAPD3, has been recently discussed in Tane et al. 2022 and Yoshida et al. 2022 respectively. Building on these already existing data, the manuscript by Cutts et al. provides molecular insight into how concerted action of NCAPH, NCAPD2 and KIF4A regulates condensin I activity.

The data overall is interesting and will be of value to the field. The presentation of the data however is not as good as it could have been. The manuscript is written in a rather confusing manner. Some conclusions are too strong and are not supported by the experiments. There are also multiple errors throughout the manuscript, both in the text and in the figures. This unfortunately makes what could have been a nice story appear rather confusing, and on some occasions this makes me doubt the validity of the data. So in my view quite some work would be needed before this manuscript is suited for publication in EMBO Journal.

Major issues:

I have a number of major concerns regarding figure 1b:

1. First, the gel appears to be a combination of multiple cut out gels, which should be indicated properly. Otherwise, the figure creates an improper assumption that all the tested conditions belong to one experiment, which presumably is not the case. There should be visible spaces between the gels and/or explanation in the text or figure legend.

The gel was included to transparently display the composition of the samples used in the assay, as many previous studies on the ATPase activity of condensin complexes do not include SDS-PAGE analysis of all samples used. However, we understand the reviewer's desire for a more stringent comparison. To address this, we have conducted additional measurements and ran a gel including all samples. We have also added a panel showing the band for the PNP enzyme used in the ATPase assay, which serves as a loading control. We have not quantified these gels and protein concentration is determined at the point of setting up the assay. The full uncropped gel has been provided as a figure supplement.

2. Related to this point, the condition with all three plusses is present three times (column 1, 3 and 8). I assume this is done because these are in fact separate gels from potentially separate preps? These three columns have quite different ATP/molecule/min values. Some variability between experiments is understandable, but when the control between all three experiments is so different, a fair comparison of other samples between the different samples, cannot be made. For example, when comparing column 1 with column 2, authors state that this is a meaningful difference. But then, column 8, which should be a control, is pretty much the same

value as column 2, which is where we should see a significant difference according to the authors. Comparing bar 1 with bar 7 is also not fair in this case.

Similar to comment 1 raised by Reviewer 1, we acknowledge that the labelling of this figure may have caused significant confusion. We have thoroughly revised the figure to ensure clarity and avoid any misunderstandings.

Samples in column 1, 3 and 8 of the original Figure are different samples. Column 1 is a purified pentamer, column 3 is a tetramer lacking NCAPD2 with separately purified NCAPD2 added, and column 8 is a tetramer lacking NCAPG with separately purified NCAPG added. We have changed the labelling to specify what complexes, subcomplex and subunits were mixed, and have provided lane references both in the figure and in the text to help guide the reader of the differences between each condition. This revised labelling should make it clear that the sample in column 1 is not the same as the sample in column 8.

With regards to the comment that sample 1, 3 and 8 should have the same measurement, this result would only be expected if we were able to completely reconstitute the pentameric complex by adding the missing HAWK to a tetramer. However, to produce a stable pentamer, all subunits need to be co-expressed prior to purification. It is common for large complexes not to be able to be fully reconstituted by mixing purified subcomplexes, and we have added explicit mention of this in the main text (Lines 125-127):

“The condensin I complex was most stable when all subunits were co-expressed, and while adding HAWK subunits to tetrameric complex partially reconstituted pentamers, complete reconstitution could not be achieved.”

We have also changed the wording to indicate that the reconstitution is incomplete and rescue is towards pentameric levels (Lines 121-125):

“The hyperactivity of tetrameric complexes was partially rescued towards pentameric levels when an excess of recombinant NCAPG or NCAPD2 were added (Fig. 1B, lanes 3 and 8), with the addition of NCAPG and NCAPD2 to tetrameric complexes resulting in partial reconstitution of pentameric complex (Fig. EV1B, C).

Despite partial reconstitution, our data is robust, we show that loss of a HAWK increases the ATPase rate compared to wild-type pentamer and this effect can be partially reversed upon partial reconstitution. This reconstitution approach acts as an internal control to ensure we are not observing variability between protein preparations and demonstrates that indeed it is the loss of the HAWK what causes the increased rate.

3. Labelling of the bands requires improvement as it is not clear which band belongs to which protein. In the first 4 lanes there is an extra band (or even two). Is this a degradation product? Also, due to the fact that there are multiple gels that have been glued together in this figure, it looks like the bands are not aligned properly, which makes for a very confusing unit. For example, in column 5 we should only see a band corresponding to NCAPD2. But this band runs at the height of SMC4 in other columns, for example 2.

The revised Fig. 1 has a gel where all samples have been run side by side.

The reviewer is correct, there was some small degradation, particularly in the CIΔD2 sample, loss of NCAPD2 appears affects complex stability. A fresh purification of this complex has significantly improved this.

To ensure purification difference did not affect our conclusions, we have included a wide range of experimental controls, including reversing the hyperactivity of the tetramer complex by adding the missing HAWK subunit back and active site mutations ensuring that the ATPase activity is derived from the condensin complex. While there are some differences between preps, we have observed that tetramers have significantly higher ATPase activity than pentamers CI, when we directly compare data collected at similar times of the two protein batches of CIWT, CIΔD2 and CIΔG used in this study (Fig. EV1A). The larger size of NCAPD2 and NCAPG is due to the addition of a strep purification tag.

4. And more importantly: in column 7, in which the NCAPG subunit should not present, the band is still there. Is this a mistake or how can this be explained?

This is indeed a labelling error, we can only apologise for not spotting this error prior to submission. The new gel used in the Fig. 1 has resolved this point.

5. All the data presented in figure 1b are potentially exciting, but until all the points mentioned above have been addressed, the figure remains confusing. These various points also make me doubt the validity of the data. If there simply is lots of experimental variation between the different wild type preps, then an equal number of preps is also needed for all the mutant preps. And it would be essential to plot the relative spread of the results both between preps and within preps, depicting the individual data points for all. And then please load all on one gel, with the correct labels for each sample.

As explained in the previous replies lanes 1, 3, and 8 in our gel were not identical samples. The proteins in these lanes were distinct, and as a result, their ATPase rates varied. To address this, we have now made improvements to our gel labelling, ensuring that all samples are more clearly identified, and we have rerun all the samples on the same gel. We have explicitly clarified this in the revised manuscript, and we trust this will resolve the confusion.

In the course of our work, we produced two independent preparations of condensin I pentamer, along with each tetramer. We are pleased to report that the ATPase rate trends were consistent between these two preparations. Specifically, we observed that the tetramers consistently exhibited higher ATPase activity than either of the pentamer preparations, in line with our original findings.

As presented in the revised Fig. EV1A, the ATPase rates of both the condensin I pentamer and the CIΔG tetramer were consistent across preparations. Additionally, we made improvements to the CIΔD2 preparation, which further enhanced the phenotypic outcome. These additional results underscore the reproducibility of our data and lend further confidence to our conclusions.

We believe that the revisions, which include clearer labelling, improved presentation of the gels, and additional data on the consistency of ATPase rates across independent preparations, will fully address the reviewer's concerns. Our goal is to demonstrate the reliability of our findings and clarify any ambiguity present in the initial version.

We are confident that the revised submission will provide a clearer and more compelling presentation of our results and better highlight the exciting potential of our findings. We hope these revisions satisfactorily address all concerns raised.

Further points on figure 1:

6. In figures 1b and 1c, the authors explain that ATP activity assays are performed. It is clear that the difference between these experiments is the presence of DNA. But these two panels also show the measurements in a different way, which would be needed to have explained to the reader.

We have provided extra explanations to the normalisation in the text (lines 139-142), in the Fig. 1 legend and the source data provided contains the data prior to normalisation.

7. With regards to figure 1c, I find it confusing that the CIΔG and CIΔD2 conditions were not saturated with DNA to the same extent as the CI condition. What would the fold stimulation of ATPase rate look like for CIΔG and CIΔD2 at 13uM?

We thought the activity of CIΔG and CIΔD2 had sufficiently plateaued at 6.4 uM and added 12.8 uM concentration of DNA for CI to reach the maximum DNA stimulation. However, to address this comment, we have collected CIΔG and CIΔD2 data with 12.8 uM of DNA.

8. In extended data figure 1b, there is a peak at about 144kDa corresponding to NCAPD2 subunit alone, a peak around 414kDa of Condensin I missing NCAPD2 subunit, and also a small peak of 544kDa of the reconstituted pentamer. First, the reconstituted pentamer is a very small fraction, only 7%, which is a bit worrying. Secondly, what does the peak at 255kDa correspond to? It represents the biggest fraction the sample.

In response to this point, we have collected the mass-photometry data again, and the size of 544 kDa appears to have been an instrumental offset. Existing smaller masses appear to be degradation products.

The new prep sample of CIΔD2 has one main peak encompassing 80% of the events at 458 kDa, where the expected MW is 486 kDa (~5% error). Upon addition of NCAPD2, two additional peaks appear; one at 160 kDa for NCAPD2 and the other at 620 kDa for the CI pentamer, encompassing 17 and 22% of events respectively, while the CIΔD2 peak, fit at 465 kDa encompasses 35% of events. This data is consistent with partial reconstitution of the pentamer (Fig. EV1C).

Despite attempts at adding more NCAPD2, complete reconstitution of the pentamer from the tetramer was not observed. We have commented on this in the text, but as stated above in order for condensin to form a stable pentamer, all subunits need to be co-expressed (Line 120-127). All subsequent mutations were produced via co-expression as a pentamer to overcome the limitations of reconstitution after purification, and resulted in one major species consistent with pentameric mass (Fig. EV1 D,E,F).

9. In figure 1d, for specifically the CI Δ D2 samples, the bands shift all the way up into the slots. Why is that? Does this Δ D2 setting cause the complex to multimerise? Please comment.

We believe that as the reviewer suggests this might be due to multimerization. We have added a comment in the text speculating that the upward shift is caused from complex multimerization (lines 153-155):

“Loss of NCAPD2 resulted in a minimal reduction in binding, and a higher shifted band, suggesting it may multimerise.”

Figure 2:

10. In figure 2a in the CI D2 Δ C, and also in the CI D2 Δ C QL sample, the C-terminus of NCAPD2 is removed. That is a deletion of 1305 to 1401 amino acid. Because of that the protein can be expected to run lower on the gel, but would it be so much lower that it overlaps with the SMC2-corresponding band? Is it possible to run the gel longer or on a different percentage gel to better separate out the proteins? Now it looks as if CI D2 Δ C didn't express the NCAPD2 subunit at all. Also, the abbreviation of CIH Δ N QL and CI Δ D2 QL is different than in the figure legends. Ideally this would be consistent, also with the Q-loop mutants in figure 1b.

We have run another gel and bands are now better resolved, confirming that the D2 subunit is present. We have also performed mass photometry on N Δ H and D2 Δ C condensin I pentamers to provide additional proof they are indeed pentameric (Fig. EV1D,E,F). We have ensured that the labelling is consistent between figures, using CIQ rather than QL.

11. In figures 1b and 2a, it would be helpful to depict individual data points as well. Are the values represented by graph bars in figure 1b a representation of multiple replicates? And what then constitutes a replicate?

We have added individual data points for all plots.

Replicates are the same experiment, repeated by defrosting a new aliquot of protein stock of each sample and setting up the plate again, ideally on a different day. Hence, while these are technical replicates, they include error involving pipetting, room temperature changes, sample handling, etc, and are standard procedure for this type of biochemical assay. Some protein stocks ran out during the course of preparing the manuscript (namely CI, CI Δ G and CI Δ D2), hence the results are averages of these two preps such that data of these samples also represents difference in protein purification of active experimental samples. Data was analysed with an unpaired, two-tailed t-test with Welch's correction to account for unequal variances and sample sizes. The number of repeats is included in the figure legends.

12. Figure 2d shows that the NCAPG subunit alone binds the NCAPD2 peptide very efficiently. Condensin I pentamer binds the peptide much less well, which the authors speculate may be due to the fact that the NCAPD2 already present within the complex subunit is occupying NCAPG, and that the fluorescently labelled peptide cannot compete with it. How do authors then explain that in figure 2g with the NCAPH peptide this does not seem to be the case? Condensin I pentamer appears to bind NCAPH peptide even more efficiently than the NCAPG subunit alone.

NCAPH peptide can then compete for binding to NCAPG with the NCAPH present within the pentamer? An note that a line connecting the data points for C1ΔG is missing in figure 2g.

FP examines the change in fluorescence polarisation signal of a rapidly tumbling peptide vs a peptide bound to a more slowly tumbling macromolecule. The extent of FP signal depends on many factors, such as molecular weight and shape. Hence, one cannot directly compare the extent of FP signal of the NCAPH in the presence of NCAPG and condensin I, as NCAPG and condensin I have different masses and shapes.

To do this comparison, saturating protein concentrations would be required to fit the K_d of the NCAPH peptide to NCAPG and condensin I, as performed for NCAPD2 peptide. Unfortunately, these complexes cannot be sufficiently concentrated to collect such data.

However, to provide some evidence that the NCAPH peptide may compete with that found in the condensin I pentamer, we have collected FP data using the C1NΔH pentamer, which has similar size and likely shape to condensin I pentamer, but lacks the NCAPH N-terminus. This resulted in higher FP signal than condensin I pentamer, suggesting competition between the fluorescent peptide and equivalent NCAPH region in the complex.

In the revision, we have added trend lines for all FP data, and used solid lines where the K_d is fit and dashed lines for when K_d fitting was not possible, to help distinguish between data where the K_d has been fit.

13. For figures such as 2d and 2g, please (also) plot the data with a log-scale axis, as this should allow for an S-shape of the curve. This benefits the K_d calculation/visualization.

We agree with the reviewer that for high affinity binding, plotting on a log scale shows off the shape of the data nicely, however, at the low affinity range, plotting on a log scale makes the curvature at higher concentration difficult to observe, we provide an example for the KIF4A binding data below. Unfortunately, we are at the solubility limit of the condensin complex components, so attaining higher concentration data points is not possible.

The calculation of the K_d is not affected by whether x-axis is on a log scale or not, as this is a mathematical minimisation. The K_d fit is displayed as a solid line, to visualise the quality of fit.

Figure 3:

14. In figure 3d, the authors use NCAPG protein with mutations D136K, D137K and D141K. These mutations were introduced in the acidic patch that was predicted to bind KIF4A by AlphaFold. This construct is used to show the abrogated binding between NCAPG and the KIF4A peptide. Could this mutant be used in binding assays with the NCAPD2 peptide? This is supposedly where NCAPD2 is predicted to bind NCAPG according to AlphaFold structures. This would further strengthen the data from figure 2d and provide an additional layer of evidence for the same binding spot on NCAPG for NCAPD2 and KIF4A.

We have added FP data of the KIF4A binding deficient mutant of NCAPG to the NCAPD2 peptide (Fig. 2D). As expected, it does not bind to NCAPD2, and followed a similar trend to C Δ G.

15. Figure 3e: The data shows that deletion of C-terminus of NCAPD2 subunit results in higher ATPase activity. The authors propose that this may be due to the fact that the NCAPG is then unoccupied, which allows for KIF4A to bind and stimulate ATPase activity of the complex. If it was simply about competition between NCAPD2 and KIF4A, why is there a higher ATPase activity in D2 Δ C also without addition of KIF4A? This difference is also present in figure 2a in C1 D2 Δ C sample. This implies that it cannot be simply about occupying the spot where the activator is binding. I believe the model can still hold true because the ATPase activation is even higher upon addition of KIF4A peptide, but it cannot be the only explanation and this should be elaborated on in the text.

Our data suggests that three SLiMs bind to NCAPG, one from NCAPD2, one from NCAPH and the third from KIF4A. Loss of NCAPD2 and/or NCAPH SLiM increased ATP turn over and we propose this is due to these interactions preventing normal conformational changes and DNA loading. Our model is that KIF4A activates condensin by displacing inhibitory interactions. We believe this model is consistent with all our data.

To provide further proof of this model we have performed a competition FP assay using the 5-FAM labelled NCAPH peptide and condensin I pentamer lacking both NCAPH and NCAPD2 peptides. The NCAPH peptide bound to this with resultant FP of ~35mP, which was significantly reduced to ~20mP on addition of the KIF4a peptide. This provides evidence of that KIF4A can to some extent compete with NCAPH (Fig 3E,F).

We also have tested the effect of the KIF4A peptide on condensin I mutants lacking the NCAPH SLiM and lacking both the NCAPH and NCAPD2 SLiMs, and found that KIF4a was not able to stimulate the ATPase rate further (Fig. 3G).

Collectively, the data strengthens the hypothesis that KIF4A competes with the NCAPH peptide, but also suggests that the main role of the NCAPD2 is to modulate KIF4A binding affinity, while NCAPH directly influences ATPase rate. Given that the loss of NCAPD2 SLiM also increased ATPase rate, binding of NCAPD2 to NCAPG could contribute to NCAPH affinity. We have included these discussion points (Lines 359-365).

We have also added an alternative hypothesis (lines 365-369), as previous studies have suggested that the release of N-terminal kleisin is a key step in the condensin and cohesin ATPase cycle (Hassler, et al, Mol Cell, 2019, Muir, et al, NSMB, 2020). An alternative model could be that NCAPH interacting with NCAPG alters the dynamics of this step.

16. Also figure 3e: I would like to ask for clarification what is plotted on the y axis. Is it the same scale as in figure 1b? The wording is different and not properly explained in the text or the figure legend. It would also be helpful to keep the labelling coherent, so 'D2ΔC' could be D2ΔC, like used previously.

The y-axis is the same scale as 1b, and we have updated this labelling and used CID2ΔC consistently figures.

Figure 5:

17. The authors base their hypothesis on 'existing models' in which the loop extrusion initiation requires DNA threaded through a compartment formed by the kleisin (discussed in lines 276-278). No literature is referenced to support these 'existing models'. This referee is not aware of any such model. Are the authors maybe referring to Lee et al, NSMB 2020? But then that paper does not include loop extrusion experiments. Or perhaps the authors meant to refer to other papers? Either way, please make sure to cite the appropriate work.

The model our figure was based on is from Shaltiel et al, Science, 2022, although with slightly simplified presentation. We have included this citation. However, a revised model has been presented by Dekker et al, 2023, Science, hence we have revised our figure (Fig. 5) to be more consistent with this model and also provided this citation.

In these models, little detail is provided as to the mechanism of DNA loading, and we are the first to use the term "DNA threading" to describe how the first step of how a loop extruding complex could be formed (lines 358-359).

18. The drawing of the model could be significantly improved, for example in panel b it is not clear that NCAPD2 and NCAPH are contacting NCAPG. There are white blocks covering the NCAPG subunit. The same can be said about the c panel and KIF4A binding. It would be helpful to label the subunits of condensin I in the figures, as well as KIF4A.

We have modified figure 5 and have included subunit labels in Fig. 5A and for KIF4A in Fig. 5C. The white shading was added to make it easier to see where the DNA and NCAPH passes, but perhaps these rendered poorly. We have now used a simpler DNA rendering, white with a black outline to provide contrast against the coloured subunits.

19. At the end of the paragraph 'Motif binding is conserved across species' the authors discuss data available in literature to strengthen their point about the importance of certain regions in Sgo1 and Lrs4. It is not clear that the data discussed here is not part of this manuscript, I would disclose this better. Also the data is not explained well at all. What is a Q325stop mutation? What is smc2-157 mutation and why is it relevant? This needs to be properly explained to the reader.

We have used the phrases "previously been shown", "previous work" and "recent work", and provided references when referring to other's work, to discriminate between our work.

The Q325stop mutation is a truncation of Lrs4 lacking part of the Ycg1/NCAPG binding motif, and this results in a fitness phenotype when used in combination with smc2-157 which has a single point mutation close to the Walker B motif nucleotide binding site, reducing activity. Combining multiple mutations to observe fitness phenotypes is a commonly employed yeast genetics approach. We have simplified

our description of this previous finding to better highlight the main conclusion; that loss of the interaction between Lrs4 and Ycg1 results in a cellular phenotype, hence is physiologically significant (Lines 345-350).

20. In the abstract, line 27, authors say that KIF4A directly competes with N-terminus of NCAPH and C-terminus of NCAPD2. The competition between KIF4A and NCAPD2 is shown in figure 3d, in which KIF4A peptide binds with higher affinity to Condensin I lacking C-terminus of NCAPD2. However, the competition between KIF4A and N-terminus of NCAPH was never shown. The same wording is used in the introduction "...we show that competition between KIF4A and NCAPD2/H binding results in condensin I activation" as well as in the title of a paragraph starting in line 213. I am afraid this was not shown in the manuscript. Appropriate experiments must be performed to draw these conclusions, or the wording in the manuscript needs to be adjusted.

We have added data strengthening the result that NCAPH competes with KIF4A, including FP competition data and ATPase assay data demonstrating that KIF4A could not stimulate condensin pentamer lacking the NCAPH peptide (Fig. 3F and G). However, we acknowledge that the competition between KIF4A and NCAPH is less clear, and have softened the language when discussing it in the text ("appears" Line 23, "potential" line 106 and 356).

Minor issues:

1. Affiliation 2 and 3 are identical

We have ensured affiliations are presented in line with EMBO formatting guidelines.

2. The authors may wish to update the information in the introduction (line 88) and in figure 7, regarding a newly identified NCAPG2 binder and condensin II regulator, M18BP1.

We have included this reference. An author on this work, E. Cutts, is also a corresponding author on the M18BP1 work.

3. The sentence starting in line 191: "The interface between the NCAPD2 SLiM and NCAPG..." is very difficult to read. It is not clear whether the mentioned set of residues belongs to NCAPD2 or NCAPG. Please clarify.

We have rewritten this sentence, and have now referred to residue numbers in subscript to make it clearer which proteins they are found in (line 213)

4. Similarly, the sentence in lines 223-227 is rather difficult to understand.

Again, we have rewritten this sentence, and have now referred to residue numbers in subscript to make it clearer which proteins they are found in.

5. Line 223 contains a typo in the word 'mediated'.

We have corrected this typo.

6. In figure 2c, the amino acids of NCAPD2 are mislabeled. The two glutamic acids (E) should not be at positions 1382 and 1384 but 1381 and 1383. The same goes for the two arginines (R). Instead of 1398 and 1399 they should be at 1397 and 1398.

We have corrected these.

7. In figure 2e, the amino acid D63 is mislabeled. At position 63 there is a glutamic acid (E).

We believe reviewer is referring to figure 2f. We have corrected this.

8. In figure 2h, the amino acids are mislabeled. It should be F1220 and F122.

our description of this previous finding to better highlight the main conclusion; that loss of the interaction between Lrs4 and Ycg1 results in a cellular phenotype, hence is physiologically significant (Lines 345-350).

20. In the abstract, line 27, authors say that KIF4A directly competes with N-terminus of NCAPH and C-terminus of NCAPD2. The competition between KIF4A and NCAPD2 is shown in figure 3d, in which KIF4A peptide binds with higher affinity to Condensin I lacking C-terminus of NCAPD2. However, the competition between KIF4A and N-terminus of NCAPH was never shown. The same wording is used in the introduction "...we show that competition between KIF4A and NCAPD2/H binding results in condensin I activation" as well as in the title of a paragraph starting in line 213. I am afraid this was not shown in the manuscript. Appropriate experiments must be performed to draw these conclusions, or the wording in the manuscript needs to be adjusted.

We have added data strengthening the result that NCAPH competes with KIF4A, including FP competition data and ATPase assay data demonstrating that KIF4A could not stimulate condensin pentamer lacking the NCAPH peptide (Fig. 3F and G). However, we acknowledge that the competition between KIF4A and NCAPH is less clear, and have softened the language when discussing it in the text ("appears" Line 23, "potential" line 106 and 356).

Minor issues:

1. Affiliation 2 and 3 are identical

We have ensured affiliations are presented in line with EMBO formatting guidelines.

2. The authors may wish to update the information in the introduction (line 88) and in figure 7, regarding a newly identified NCAPG2 binder and condensin II regulator, M18BP1.

We have included this reference. An author on this work, E. Cutts, is also a corresponding author on the M18BP1 work.

3. The sentence starting in line 191: "The interface between the NCAPD2 SLiM and NCAPG..." is very difficult to read. It is not clear whether the mentioned set of residues belongs to NCAPD2 or NCAPG. Please clarify.

We have rewritten this sentence, and have now referred to residue numbers in subscript to make it clearer which proteins they are found in (line 213)

4. Similarly, the sentence in lines 223-227 is rather difficult to understand.

Again, we have rewritten this sentence, and have now referred to residue numbers in subscript to make it clearer which proteins they are found in.

5. Line 223 contains a typo in the word 'mediated'.

We have corrected this typo.

6. In figure 2c, the amino acids of NCAPD2 are mislabeled. The two glutamic acids (E) should not be at positions 1382 and 1384 but 1381 and 1383. The same goes for the two arginines (R). Instead of 1398 and 1399 they should be at 1397 and 1398.

We have corrected these.

7. In figure 2e, the amino acid D63 is mislabeled. At position 63 there is a glutamic acid (E).

We believe reviewer is referring to figure 2f. We have corrected this.

8. In figure 2h, the amino acids are mislabeled. It should be F1220 and F122.

We believe reviewer is referring to figure 2c. We have corrected such that it displays F1220 and F1221.

9. Is the color-coding in figure 3g not correct? Should the 'down-size' be red and 'up-size' blue? If so, then please have this corrected, or else please better explain? Indeed, the colouring between 3G and 3D did not match, we have corrected this such that it is consistent.

Referee #3:

The condensin complexes compact mitotic chromosomes through loop extrusion. Whether and how the loop extrusion activities of condensin are regulated are not well understood. Using in vitro biochemical and single-molecule biophysical experiments, Cutts et al. show that the ATPase activity and loop extrusion of human condensin I can be activated by a small motif in the kinesin KIF4A. They also show that yeast condensin interactors, such as Lrs4 and Sgo1, contain a similar motif, suggesting that this motif is conserved.

Although the interaction between condensin I and KIF4A has been reported previously, the biochemical function of this interaction is unknown. The current study provides strong evidence to indicate that KIF4A directly regulates the biochemical activity of condensin I and suggests a potential mechanism. As such, it will be of great interest to scientists in the chromosome biology field and should be published.

I have the following suggestions that the authors may wish to address prior to publication.

Major points

(1) I found it surprising that condensin I lacking NCAPG was still capable of loop extrusion. Is this consistent with data in human cells? Specifically, do cells lacking NCAPG exhibit partial chromosome condensation? In addition, how does the current models of loop extrusion by condensin explain this finding?

We have added the following discussion (lines 413-423):

“Knockdown of both NCAPG and NCAPG2 results in loss of chromosome compaction, suggesting condensin tetramers are not active in human cells (Ono et al, 2003). This suggests an essential role for NCAPG, further demonstrated by its embryonic lethality when knocked out (Sun et al, 2022). Our data suggests NCAPG could play an important role as an interaction scaffold enabling site specific localisation and regulation, and recent work has found that mutation of the SLiM NCAPH58-68 which binds NCAPG affects cellular fitness (Ambjørn et al, 2024). However, condensin tetramers are thought to exist in some eukaryotes, as NCAPG2 gene is absent in Drosophila melanogaster (King et al, 2019) and previous studies demonstrate tetrameric condensin complex can extrude DNA loops but are prone to slippage (Shaltiel et al, 2022).

And we have added a section regarding the tetrameric activity to our speculative model (Fig. 5D).

(2) In Fig. 2g, the NCAPG-binding motif of NCAPH binds to free NCAPG alone and the intact condensin I with similar affinity. This suggests that this motif of NCAPH

does not bind to NCAPG with sufficient occupancy in intact condensin in the ground state. The authors should discuss this discrepancy and its implications for their proposed autoinhibition model.

Addressed via reviewer 2, comment 12, copied below:

FP examines the change in fluorescence polarisation signal of a rapidly tumbling peptide vs a peptide bound to a more slowly tumbling macromolecule. The extent of FP signal depends on many factors, such as molecular weight and shape. Hence, one cannot directly compare the extent of FP signal of the NCAPH in the presence of NCAPG and condensin I, as NCAPG and condensin I have different masses and shapes.

To do this comparison, saturating protein concentrations would be required to fit the K_d of the NCAPH peptide to NCAPG and condensin I, as performed for NCAPD2 peptide. Unfortunately, these complexes cannot be sufficiently concentrated to collect such data.

However, to provide some evidence that the NCAPH peptide may compete with that found in the condensin I pentamer, we collected FP data using the CIN Δ H pentamer, which has similar size and likely shape to as condensin I pentamer, but lacks the NCAPH N-terminus. This results in higher FP signal than condensin I pentamer, suggesting competition between the fluorescent peptide and equivalent NCAPH region in the complex.

(3) The model presented in Fig. 5 is speculative and adds little to the manuscript. Figure 5 provides a visual illustration of our model, we have included wording such as “model”, “suggest”, “could”, “potential” (lines 310-314, 356, 361, etc) to help to convey the speculative nature of the model. We feel the illustrated model helps us discuss our findings in a broader loop-extrusion context. We have further added to the utility of figure 5 by adding details as to how tetrameric loop extrusion can occur (Fig. 5D), to better address the first point raised.

(4) Fig. 6 and Fig. 7 can be combined. Better still, they should test whether Sgo1 or Lrs4 binding stimulates the loop extrusion and/or ATPase activity of the yeast condensin.

We have combined figure 6 and 7.

This study demonstrates human condensin I is auto inhibition and these interactions can be displaced by KIF4a to activate the complex. These auto-inhibitory regions, are absent in yeast (EVFig. 3A,B), hence there is no reason to suspect that Sgo1 or Lrs4 peptides should alter yeast condensin loop extrusion rates. Sgo1 and Lrs4 are more likely localisation factors, helping to recruit or stall yeast condensin at specific sites, which has been demonstrated *in vivo* for Sgo1 (Wang, et al, Biorxiv, 2024). To observe this effect *in vitro* experiments would likely need the larger complexes Sgo1 and Lrs4 form and pericentromeric DNA and rDNA, respectively, which we feel is beyond the scope of this work. This is, however an exciting question, which should be explore in the future, and we have included a comment to this effect in the text (lines 440-443):

“Whether yeast condensin SLiMs also regulate DNA loop extrusion is an open question, and should be investigated in future in the context of the larger complexes formed by Sgo1 and Lrs4 and the regions of pericentromeric and rDNA they act upon.”

Prof. Luís Aragón
MRC Laboratory of Medical Sciences
Du Cane Road
London, London W12 0HS
United Kingdom

23rd Nov 2024

Re: EMBOJ-2024-117957R
Molecular Mechanism of Condensin I Activation by KIF4A

Dear Luis and Erin,

Thank you for submitting your revised manuscript to The EMBO Journal. Two of the original referees have now assessed it once again (see comments below), and overall consider the study satisfactorily improved and in principle ready for publication. Nevertheless, they both ask for some further textual modifications, which I would appreciate if you could incorporate during a final round of minor revision. In addition, there are also several editorial issues to be taken care of at this point:

- Please adjust the order of the manuscript sections: Title page with complete author information, Abstract, Keywords, Introduction, Results, Discussion, Methods, Data Availability, Acknowledgements, Disclosure and Competing Interests Statement, References, Main Figure Legends, Tables, Expanded Figure Legends.
- On the abstract page of the manuscript, please include 4-5 general keyword terms to enhance searchability.
- Please double-check to make sure to all relevant funding information mentioned in the manuscript is also entered into our submission system (currently missing: Max Planck Society; IMPRS on Cellular Biophysics)
- Please adjust the format for citation of preprints as specified in our author guidelines:
The citation in the text should be: "(preprint: NAME1 et al, YEAR)"
The citation in the reference list: "Author NAME1, Author NAME2, ... (YEAR) article title. bioRxiv/ResearchSquare doi: XXX"
For the study submitted back-to-back with yours, maybe consider coordinating with the other group so that the final paper instead of the preprint could be cited.
- As we are switching from a free-text author contribution statement towards a more formal statement based on Contributor Role Taxonomy (CRediT) terms, please remove the present Author Contribution section and instead specify each author's contribution(s) directly in the Author Information page of our submission system during upload of the final manuscript. See <https://casrai.org/credit/> for more information.
- Please rename the Conflict of Interest section into "Disclosure and Competing Interests Statement", in accordance with our updated Guide to Authors (<https://www.embopress.org/competing-interests>)
- Please cut the EV movie legends from the main text, instead placing each one into one separate legend text file per EV movie; then move each legend file together with the respective movie file into a separate ZIP archive before re-uploading as "Movie EV1/2/3..."
- Please note that source data files need to be saved in a scheme, one figure/folder, and then uploaded as .zip files. E.g. all the Source data files for figure 1 need to be saved in a single folder and this needs to be zipped and then uploaded as "SD figure 1.zip" file. For EV and/or appendix figures, please combine all source data in one single ZIP.
- Please provide suggestions for a short 'blurb' text prefacing and summing up the study in two sentences (max. 250 characters), followed by 3-5 one-sentence 'bullet points' with brief factual statements of key results of the paper; they will form the basis of an editor-written 'Synopsis' accompanying the online version of the article. Please also upload a synopsis image, which can be used as a "visual title" for the synopsis section of your paper. The image should be in PNG or JPG format with the modest dimensions of EXACTLY 550 pixels wide and 300-600 pixels high (maybe based on a Figure 6F?).
- Finally, during routine pre-acceptance checks, our data editors have raised the following queries regarding figures, data, and legends; I would appreciate if you briefly answered to them in the cover letter of your final submission, and made the requested text modifications with changes/additions highlighted via the "Track changes" option, to facilitate our final checking.
 1. Please note that the legend for figure 6F is missing in the manuscript. This needs to be rectified.
 2. Please note that the exact p values are not provided in the legends of figures 3E; 4B, C, G.
 3. Please indicate the statistical test used for data analysis in the legends of figures 3E, 4B, C.

4. Please note that the box plots need to be defined in terms of minima, maxima in the legend of figure 4G.
5. Please note that information related to n is missing in the legends of figures 3D, E; 4G, 6D, E.
6. Although 'n' is provided, please describe the nature of entity for 'n' in the legend of figure 1C.
7. Please note that the error bars are not defined in the legend of figure 3E.
8. Please note that the measure of center for the error bars needs to be defined in the legends of figures 2D, G; 3D, F; 6D, E.

I am therefore returning the study to you once more, to allow you to incorporate these final changes and upload all requested files. Once we have received them, we should be able to swiftly proceed with acceptance and production of the manuscript.

With kind regards,

Hartmut

- 1) Every manuscript requires a Data Availability section (even if only stating that no deposited datasets are included). Primary datasets or computer code produced in the current study have to be deposited in appropriate public repositories prior to resubmission, and reviewer access details provided in case that public access is not yet allowed. Further information: embopress.org/page/journal/14602075/authorguide#dataavailability
- 2) Each figure legend must specify
 - size of the scale bars that are mandatory for all micrograph panels
 - the statistical test used to generate error bars and P-values
 - the type error bars (e.g., S.E.M., S.D.)
 - the number (n) and nature (biological or technical replicate) of independent experiments underlying each data point
 - Figures may not include error bars for experiments with $n < 3$; scatter plots showing individual data points should be used instead.
- 3) Revised manuscript text (including main tables, and figure legends for main and EV figures) has to be submitted as editable text file (e.g., .docx format). We encourage highlighting of changes (e.g., via text color) for the referees' reference.
- 4) Each main and each Expanded View (EV) figure should be uploaded as individual production-quality files (preferably in .eps, .tif, .jpg formats). For suggestions on figure preparation/layout, please refer to our Figure Preparation Guidelines: <http://bit.ly/EMBOPressFigurePreparationGuideline>
- 5) Point-by-point response letters should include the original referee comments in full together with your detailed responses to them (and to specific editor requests if applicable), and also be uploaded as editable (e.g., .docx) text files.
- 6) Please complete our Author Checklist, and make sure that information entered into the checklist is also reflected in the manuscript; the checklist will be available to readers as part of the Review Process File. A download link is found at the top of our Guide to Authors: embopress.org/page/journal/14602075/authorguide
- 7) All authors listed as (co-)corresponding need to deposit, in their respective author profiles in our submission system, a unique ORCID identifier linked to their name. Please see our Guide to Authors for detailed instructions.
- 8) Please note that supplementary information at EMBO Press has been superseded by the 'Expanded View' for inclusion of additional figures, tables, movies or datasets; with up to five EV Figures being typeset and directly accessible in the HTML version of the article. For details and guidance, please refer to: embopress.org/page/journal/14602075/authorguide#expandedview
- 9) To facilitate reproducibility and cross-laboratory adoption of methodologies, please structure the Materials & Methods section as outlined in our guide to authors, including a completed Reagents and Tools Table that can be downloaded from our author guidelines as well (<https://www.embopress.org/page/journal/14602075/authorguide#structuredmethods>).
- 10) Digital image enhancement is acceptable practice, as long as it accurately represents the original data and conforms to community standards. If a figure has been subjected to significant electronic manipulation, this must be clearly noted in the figure

legend and/or the 'Materials and Methods' section. The editors reserve the right to request original versions of figures and the original images that were used to assemble the figure. Finally, we generally encourage uploading of numerical as well as gel/blot image source data; for details see: embopress.org/page/journal/14602075/authorguide#sourcedata

At EMBO Press, we ask authors to provide source data for the main manuscript figures. Our source data coordinator will contact you to discuss which figure panels we would need source data for and will also provide you with helpful tips on how to upload and organize the files.

In the interest of ensuring the conceptual advance provided by the work, we recommend submitting a revision within 3 months (21st Feb 2025). Please discuss the revision progress ahead of this time with the editor if you require more time to complete the revisions. Use the link below to submit your revision:

Link Not Available

Referee #1:

It has been a long-standing question regarding how the loop extrusion activity of condensin is controlled in the cell cycle. This study carried out elegant experiments to address the molecular mechanism by which condensin I is activated by the chromokinesin, KIF4A. Cutts E. et al. showed that KIF5A directly interacts with NCAPG, the HAWK subunit of condensin I, using a conserved motif called SLiM found in its C-terminal tail. Interestingly, this interaction is directly competed by the similar SLiMs in NCAPH and NCAPD2, two other subunits of condensin I. Importantly, the SLiM-containing KIF4A peptide is sufficient to stimulate condensin I ATPase and DNA loop extrusion activity in vitro. The experiments are appropriately designed and performed, with the data properly interpreted. The overall conclusion of this study is supported from the data. The writing of the manuscript has been substantially improved. Below are the suggestions for the authors to consider.

1. In lines 321-325 of the Results section, the authors state that "The equivalent subunits in condensin II and cohesin, NCAPG2 and STAG1/2, respectively, have also previously been shown to act as a scaffold for SLiM interactions (Houlard et al, 2021; Li et al, 2018; García-Nieto et al, 2023; Borsellini et al, 2024)", I would suggest to include the citation of three relevant literatures here (PMID: 38714893 for the FDF motif-containing SLiM in CENP-U; PMID: 39110738 and PMID: 19696148 for the FGF motif-containing SLiM in Wapl).
2. Similarly, in lines 448-452 of the Discussion section, the authors describe that "In cohesin, the NCAPG equivalent subunit, STAG1/2, is bound by SLiMs found in WAPL, promoting cohesin unloading, CTCF, resulting in stalling/localisation, CENP-U, promoting centromere cohesion, and SGO1, protecting centromeric cohesin (Li et al, 2020; García-Nieto et al, 2023; Yan et al, 2024)", I would suggest to include the citation of three relevant literatures here (PMID: 39110738 and PMID: 19696148; for the FGF motif-containing SLiM found in Wapl).
3. In the working model shown in Figure 6F, CENP-U should be displayed both in the "Localization" and "Protection", according to reported function of CENP-U in protecting centromeric Cohesin by antagonizing Wapl (PMID: 38714893).

Referee #2:

The manuscript has improved greatly since the initial submission. The authors have addressed all the major concerns. The remaining points are all minor and textual.

The text explaining panel 3G remains unclear. Please make clearer (lines 278 to 281) that KIF4A not simply needs the N-terminus of NCAPH to activate condensin I, but that this N-terminus inhibits condensin I which is counteracted by KIF4A.

The current draft also includes other unclear sentences/typos. See e.g. the sentence at lines 363-365. So please run carefully through the entire manuscript.

In fig. 6F it is unclear what the authors mean by 'higher order regulation'. Do they mean DNA structure or protein structure or something else?

Referee #1:

It has been a long-standing question regarding how the loop extrusion activity of condensin is controlled in the cell cycle. This study carried out elegant experiments to address the molecular mechanism by which condensin I is activated by the chromokinesin, KIF4A. Cutts E. et al. showed that KIF5A directly interacts with NCAPG, the HAWK subunit of condensin I, using a conserved motif called SLiM found in its C-terminal tail. Interestingly, this interaction is directly competed by the similar SLiMs in NCAPH and NCAPD2, two other subunits of condensin I. Importantly, the SLiM-containing KIF4A peptide is sufficient to stimulate condensin I ATPase and DNA loop extrusion activity *in vitro*. The experiments are appropriately designed and performed, with the data properly interpreted. The overall conclusion of this study is supported from the data. The writing of the manuscript has been substantially improved. Below are the suggestions for the authors to consider.

1. In lines 321-325 of the Results section, the authors state that "The equivalent subunits in condensin II and cohesin, NCAPG2 and STAG1/2, respectively, have also previously been shown to act as a scaffold for SLiM interactions (Houlard et al, 2021; Li et al, 2018; García-Nieto et al, 2023; Borsellini et al, 2024)", I would suggest to include the citation of three relevant literatures here (PMID: 38714893 for the FDF motif-containing SLiM in CENP-U; PMID: 39110738 and PMID: 19696148 for the FGF motif-containing SLiM in Wapl).

These references have been added.

2. Similarly, in lines 448-452 of the Discussion section, the authors describe that "In cohesin, the NCAPG equivalent subunit, STAG1/2, is bound by SLiMs found in WAPL, promoting cohesin unloading, CTCF, resulting in stalling/localisation, CENP-U, promoting centromere cohesion, and SGO1, protecting centromeric cohesin (Li et al, 2020; García-Nieto et al, 2023; Yan et al, 2024)", I would suggest to include the citation of three relevant literatures here (PMID: 39110738 and PMID: 19696148; for the FGF motif-containing SLiM found in Wapl).

These references have been added.

3. In the working model shown in Figure 6F, CENP-U should be displayed both in the "Localization" and "Protection", according to reported function of CENP-U in protecting centromeric Cohesin by antagonizing Wapl (PMID: 38714893). This has been added to figure 6F.

Referee #2:

The manuscript has improved greatly since the initial submission. The authors have addressed all the major concerns. The remaining points are all minor and textual.

The text explaining panel 3G remains unclear. Please make clearer (lines 278 to 281) that KIF4A not simply needs the N-terminus of NCAPH to activate condensin I, but that this N-terminus inhibits condensin I which is counteracted by KIF4A.

This has been rewritten:

"We found no additional stimulation on ATPase rate when adding KIF4A₁₂₀₆₋₁₂₂₈ to either CINΔH or CINΔH,D₃ΔC, suggesting that KIF4A activates condensin I by competing and reducing binding of the inhibitory N-terminal region of NCAPH. Collectively, this suggests a model where NCAPD2 alters KIF4A binding affinity, and KIF4A competes with NCAPH to enhance ATPase activity."

The current draft also includes other unclear sentences/typos. See e.g. the sentence at lines 363-365. So please run carefully through the entire manuscript.

This has been rewritten:

"An alternative hypothesis is that interaction between NCAPH and NCAPG alters binding of the N-terminus of NCAPH to SMC2 neck. Release of the N-terminal Kleisin from the SMC neck has been shown to occur during the ATPase cycle in yeast condensin and cohesin (Hassler *et al*, 2019; Muir *et al*, 2020), hence altering the stability of this interaction may affect ATPase rate."

We have reread the manuscript and corrected typos and rephrased long sentences.

In fig. 6F it is unclear what the authors mean by 'higher order regulation'. Do they mean DNA structure or protein structure or something else?

Additional information has been provided in figure 6F as to the specifics of higher order regulation.

Prof. Luís Aragón
MRC Laboratory of Medical Sciences
Du Cane Road
London, London W12 0HS
United Kingdom

6th Dec 2024

Re: EMBOJ-2024-117957R1
Molecular Mechanism of Condensin I Activation by KIF4A

Dear Prof. Aragón,

Thank you for submitting your final revised manuscript for our consideration. I am pleased to inform you that we have now accepted it for publication in The EMBO Journal.

Yours sincerely,

Hartmut Vodermaier
